# Reconfigurable halide perovskite nanocrystal memristors for neuromorphic computing

Rohit Abraham John [1,2,8✉], Yiğit Demirağ[3,8], Yevhen Shynkarenko[1,2], Yuliia Berezovska[1,2], Natacha Ohannessian [1,4], Melika Payvand [3], Peng Zeng [5], Maryna I. Bodnarchuk [1,2], Frank Krumeich [1], Gökhan Kara [2], Ivan Shorubalko [2], Manu V. Nair[6], Graham A. Cooke[7], Thomas Lippert [1,4], Giacomo Indiveri [3✉] & Maksym V. Kovalenko [1,2✉]

Many in-memory computing frameworks demand electronic devices with specific switching characteristics to achieve the desired level of computational complexity. Existing memristive devices cannot be reconfigured to meet the diverse volatile and non-volatile switching requirements, and hence rely on tailored material designs specific to the targeted application, limiting their universality. "Reconfigurable memristors" that combine both ionic diffusive and drift mechanisms could address these limitations, but they remain elusive. Here we present a reconfigurable halide perovskite nanocrystal memristor that achieves on-demand switching between diffusive/volatile and drift/non-volatile modes by controllable electrochemical reactions. Judicious selection of the perovskite nanocrystals and organic capping ligands enable state-of-the-art endurance performances in both modes – volatile ($2 \times 10^6$ cycles) and non-volatile ($5.6 \times 10^3$ cycles). We demonstrate the relevance of such proof-of-concept perovskite devices on a benchmark reservoir network with volatile recurrent and non-volatile readout layers based on 19,900 measurements across 25 dynamically-configured devices.

[1] Department of Chemistry and Applied Biosciences, Institute of Inorganic Chemistry, ETH Zürich, CH-8093 Zürich, Switzerland. [2] Empa-Swiss Federal Laboratories for Materials Science and Technology, CH-8600 Dübendorf, Switzerland. [3] Institute of Neuroinformatics, University of Zurich and ETH Zurich, Zurich 8057, Switzerland. [4] Laboratory for Multiscale Materials Experiments, Paul Scherrer Institute, 5232 Villigen PSI, Switzerland. [5] ETH Zürich, The Scientific Center for Optical and Electron Microscopy (ScopeM), CH 8093 Zürich, Switzerland. [6] Synthara AG, Dammstrasse 16, 6300 Zug, Switzerland. [7] Hiden Analytical Ltd, Warrington WA5 7UN, UK. [8] These authors contributed equally: Rohit Abraham John, Yiğit Demirağ. ✉email: rohjohn@ethz.ch; giacomo@ini.uzh.ch; mvkovalenko@ethz.ch

The human brain operating at petaflops consumes less than 20 W, setting a precedent for scientists that real-time, ultralow-power data processing in a small volume is possible. Inspired by the human brain, the field of neuromorphic computing attempts to emulate various computational principles of the biological substrate by engineering unique materials[1–3] and circuits[4–6]. In the context of hardware implementation of neural networks, the discovery of memristors has been one of the main driving forces for highly efficient in-memory realizations of synaptic operations. Similar to evolution optimizing neurons and synapses by exploiting stable and metastable molecular dynamics[7], memristive devices of various physical mechanisms[8–10] have been discovered and developed with different volatile and non-volatile specifications. Since their inception, memristors have been utilized to implement a wide gamut of applications[11] such as stochastic computing[12], hyperdimensional computing[13], spiking[14] and artificial neural networks[15]. However, many of these frameworks often demand very different hardware specifications[16] (Fig. 1a). To meet these specifications, the memristor fabrication processes are often tediously engineered to reflect the requirements of targeted neural network configurations (e.g., neural encoding, synaptic precision, etc.). For example, the latest state-of-the-art spiking neural network (SNN) models[17,18] require memory elements operating at multiple timescales, with both volatile and non-volatile properties (from tens of milliseconds to hours)[19]. The current approach of optimizing memristive devices to a single requirement hinders the possibility of implementing multiple computational primitives in neural networks and precludes their monolithic integration on the same hardware substrate.

In this regard, the realization of drift and diffusive memristors have garnered significant attention. Drift memristors portraying non-volatile memory characteristics are typically designed using oxide dielectric materials with a soft-breakdown behaviour. In combination with inert electrodes, the switching mechanism is determined by filaments of oxygen vacancies (valence change memory); whereas implementations with reactive electrodes rely on electrochemical reactions to form conductive bridges (electrochemical metallization memory)[20]. Such drift-based memristors fit well for emulating synaptic weights that stay stable between weight updates. In contrast, diffusive memristors are often built with precisely embedded clusters of metallic ions with low diffusion activation energy within a dielectric matrix[10]. The large availability of such mobile ionic species and their low diffusion activation energy facilitate spontaneous relaxation to the insulating state upon removing power, resulting in volatile threshold switching. Memristive devices with such short-term volatility, are better suited to process temporally-encoded input patterns[21]. Hence, the application determines the type of volatility, bit-precision or endurance of the memristors, which are then heavily tailored by tedious material design strategies to meet these demands[16]. For example, deep neural network (DNN) inference workloads require linear conductance response over a wide dynamic range for optimal weight update and minimum noise for gradient calculation[15,22,23]. Whereas SNNs often demand richer and multiple synaptic dynamics simultaneously e.g., short term conductance decay (to implement synaptic cleft phenomena such as $Ca^{2+}$-dependent short-term plasticity (STP) and CAMKII-related eligibility traces[24]), non-volatile device states (to represent synaptic efficacy) and a probabilistic nature (to mimic synaptic vesicle releases[21]) (Fig. 1a). However, optimizing the active memristive material for each of these features limits their feasibility to suit a wide range of computational frameworks and ultimately increases the system complexity for most demanding applications. Moreover, these diverse specifications cannot always be implemented by combining different types of memristors on a monolithic circuit e.g., volatile and non-volatile, binary and analog, due to the incompatibility of the fabrication processes. Therefore, the lack of universality of memristors that realize not only one, but diverse computational primitives is an unsolved challenge today.

A reconfigurable memristive computing substrate that allows active control over their ionic diffusive and drift dynamics can offer a viable unifying solution but is hitherto undemonstrated. Although dual-functional memory behaviour has been observed previously, the dominance of one of the mechanisms often results in poor switching performance for either one or both modes, limiting the employability of such devices for demanding applications[25,26]. To the best of our knowledge, there is no report yet of a reconfigurable memristive material that can portray both volatile diffusive and multi-state non-volatile drift kinetics, exhibit facile switching between these two modes, and still pertain excellent performance.

Here we report a reconfigurable memristor computing substrate based on halide perovskite nanocrystals that achieves on-demand switching between volatile and non-volatile modes by

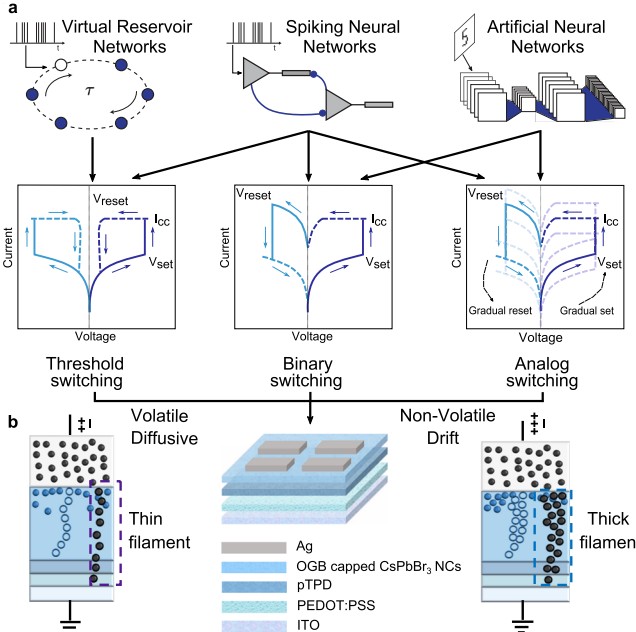

**Fig. 1 The reconfigurable perovskite memristor concept. a** Different neural network frameworks demand particular switching characteristics from in-memory computing implementations. For example, delay systems[53] (dynamical non-linear systems with delayed feedback such as virtual reservoir networks), should exhibit only a fading memory to process the inputs from the recent past. Such short-term dynamics are best represented by volatile threshold-switching memristors[46]. SNNs often demand both volatile and non-volatile dynamics, simultaneously. Synaptic mechanisms requiring STP and eligibility traces[65] can be implemented using volatile memristors[66,67] whereas synaptic efficacy requires either efficient binary-switching[68] or analog switching devices. Lastly, ANN performances specifically benefit from non-volatile features such as multi-level bit precision of weights and linear conductance response during the training phase[22,23]. **b** A reconfigurable memristor with active control over its diffusive and drift dynamics may be a feasible unifying solution. Schematic of the reconfigurable halide perovskite nanocrystal memristor is shown for reference. We utilize the same active switching material ($CsPbBr_3$ NCs capped with OGB ligands) to implement two distinct types of computation in the RC framework. The volatile diffusive mode exhibiting short-term memory is utilized as the reservoir layer while the non-volatile drift mode exhibiting long-term memory serves as the readout layer.

encompassing both diffusive and drift kinetics (Fig. 1b). Halide perovskites are newcomer optoelectronic semiconducting materials that have enabled state-of-the-art solar cells[27], solid state light emitters[28,29] and photodetectors[30–32]. Recently, these materials have attracted significant attention as memory elements due to their rich variety of charge transport physics that supports memristive switching, such as modulatable ion migration[33–35], electrochemical metallization reactions with metal electrodes[36] and localized interfacial doping with charge transport layers[37]. While most reports are based on thin films or bulk crystals of halide perovskites[33–35,38], interestingly perovskite nanocrystal (NC)-based formulations have been much less investigated till date[24,39]. NCs in general are recently garnering significant attention for artificial synaptic implementations because they support a wide range of switching physics such as trapping and release of photogenerated carriers at dangling bonds over a broad spectral region[40], and single-electron tunnelling[41]. They allow low-energy (< fJ), high-speed (MHz) operation, and can support scalable and CMOS-compatible fabrication processes. In the case of perovskite NCs, however, existing implementations often utilize NCs only as a charge trapping medium to modulate the resistance states of another semiconductor, in flash-like configurations a.k.a synaptic transistor[42–45]. The memristive switching capabilities and limits of the perovskite NC active matrix remains unaddressed, entailing significant research in this direction. Colloids of perovskite nanocrystals (NCs) are readily processable into thin-film NC solids and they offer a modular approach to impart mesoscale structures and electronic interfaces, tunable by adjusting the NC composition, size and surface ligand capping.

Our device comprises all-inorganic cesium lead bromide ($CsPbBr_3$) NCs capped with organic ligands as the active switching matrix and silver (Ag) as the active electrode. The design principle for realizing reconfigurable memristors revolves around two main factors. (i) From a material selection perspective, the low activation energy of migration of $Ag^+$ and $Br^-$ allows easy formation of conductive filaments. The soft lattice of the halide perovskite NCs facilitates diffusion of the mobile ions. Moreover, the organic capping ligands help regulate the extent of electrochemical reactions, resulting in high endurance and good reconfigurability. (ii) From a peripheral circuit design perspective, active control of the compliance control ($I_{cc}$) determines the magnitude of flux of the mobile ionic species and in turn allows facile switching between volatile diffusive and multi-bit non-volatile drift modes of operation.

The surface capping ligands are observed to play a vital role in determining the switching characteristics and endurance performance. $CsPbBr_3$ NCs capped with didodecyldimethylammonium bromide (DDAB) ligands display poor switching performance in both volatile (10 cycles) and non-volatile (50 cycles) modes, whereas NCs capped with oleylguanidinium bromide (OGB) ligands exhibit record-high endurance performances in both volatile (2 million cycles) as well as non-volatile switching (5655 cycles) modes[37,46,47].

To validate our approach and demonstrate the advantages of such reconfigurable memristive materials, we use a benchmark model of a fully-memristive reservoir computing (RC) framework interfaced to an artificial neural network (ANN)[46]. The reservoir is modelled as a network of recurrently-connected units whose dynamics act as short-term memory. Any temporal signal entering the reservoir is subject to a high-dimensional nonlinear transformation that enhances the separability of its temporal features. A linear read-out ANN layer is then connected to the reservoir units with all-to-all connections and trained to perform classification based on the temporal information maintained in the reservoir. Our RC implementation comprises perovskite memristors that are configured as diffusion-based volatile dynamic elements to implement the reservoir nodes and as drift-based non-volatile weights to implement the readout ANN layer. In their diffusive mode, the low activation energy of ion migration of the mobile ionic species ($Ag^+$ and $Br^-$) enables volatile threshold switching. The resulting short-term dynamics are essential for capturing temporal correlations within the input data stream. In the drift mode, stable conductive filaments formed by the drift of the ionic species facilitate programming of non-volatile synaptic weights in the readout layer for both training and inference. Furthermore, the readout layer can be trained online via active regulation of the compliance current ($I_{cc}$) which allows precise selection of the drift dynamics and enables multiple-bit resolution in the low resistive state (LRS). Using neural firing patterns, we show via both experiments and simulations that a RC framework based on reconfigurable perovskite memristors can accurately extract features in the temporal signals and classify firing patterns.

## Results

**Diffusive mode of the perovskite reconfigurable memristor.** We investigate two systems for diffusive dynamics- didodecyldimethylammonium bromide (DDAB) and oleylguanidinium bromide (OGB)-capped $CsPbBr_3$ NCs. The device structure comprises indium tin oxide (ITO), poly(3,4-ethylenedioxythiophene) polystyrene sulfonate (PEDOT:PSS), poly(N,N'-bis-4-butylphenyl-N,N'-bisphenyl)benzidine (polyTPD), $CsPbBr_3$ NCs and Ag as shown in Figs. 2, 3 and Supplementary Notes 1–2, Supplementary Figs. 1–3 (see "Methods" section). With a compliance current ($I_{cc}$) of 1 μA, both material systems portray volatile threshold switching characteristics with diffusive dynamics and spontaneous relaxation back to the initial state, albeit with contrasting endurance. The DDAB-capped perovskite NCs exhibit a poor on-off ratio (volatile memory a.k.a. VM $I_{power\ ON}$/ $I_{power\ OFF}$ ~ 10) and quick transition to a non-volatile state, resulting in an inferior volatile endurance of ~ 10 cycles (Supplementary Note 2, Supplementary Fig. 3). On the other hand, the OGB-capped perovskite NCs depict a highly robust threshold switching behaviour with sub-1 V set voltages, VM $I_{power\ ON}$/ $I_{power\ OFF}$ ~ $10^3$ and a record volatile endurance of $2 \times 10^6$ cycles (Fig. 3a). The volatile threshold switching behaviour can be attributed to the redistribution of $Ag^+$ and $Br^-$ ions under an applied electric field, and their back-diffusion upon removing power (Fig. 2a, Supplementary Note 2, Supplementary Fig. 4)[35,48,49]. It is also important to note that both these devices exhibit a unidirectional DC threshold switching behaviour (Supplementary Note 2, Supplementary Fig. 5) with no switching occurring under reverse bias (negative voltage on the Ag electrode). This can be correlated to the dominant bipolar electrode effect over thermal-driven diffusion, in alignment with literature[50–52].

**Drift mode of the perovskite reconfigurable memristor.** Upon increasing the $I_{cc}$ to 1 mA, both the DDAB and OGB-capped $CsPbBr_3$ NC memristors portray typical non-volatile bipolar resistive switching characteristics, once again with contrasting endurance (Figs. 2, 3, Supplementary Note 2, Supplementary Fig. 6). Both systems depict forming-free operations and similar on-off ratios (≥$10^3$). However, the DDAB-capped perovskite NCs quickly transit to a non-erasable non-volatile state, resulting in an inferior non-volatile endurance of ~50 cycles (Supplementary Note 2, Supplementary Fig. 7). On the other hand, the OGB-capped perovskite NC-based memristor portrays a highly robust switching behaviour with sub-1 V set voltages, and record-high non-volatile endurance and retention of 5655 cycles and $10^5$ s, respectively (Fig. 3b, Supplementary Note 2, Supplementary Fig. 8). Similar to the volatile threshold switching mechanism, the non-volatile resistive switching can also be attributed to

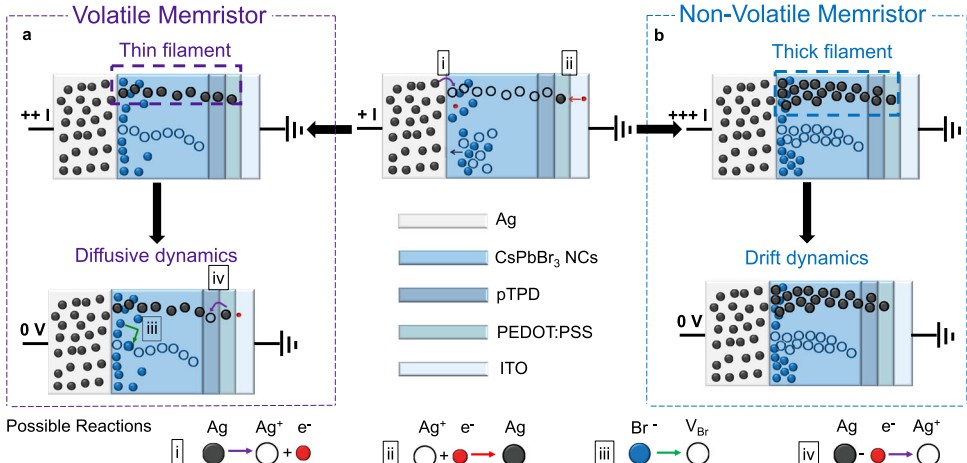

**Fig. 2 Reconfigurable halide perovskite memristor.** The device structure comprises ITO (100 nm), PEDOT:PSS (30 nm), polyTPD (20 nm), OGB-capped $CsPbBr_3$ NCs (20 nm) and Ag (150 nm). **a** Diffusive mode- illustration of the proposed volatile diffusive switching mechanism. **b** Drift mode- illustration of the proposed non-volatile drift switching mechanism. Additional note: The thickness of the individual layers in the device schematic are not drawn to scale to match the experimentally-measured thicknesses. The perovskite layer is not a bulk semiconductor, but 1–2 layers of nanocrystals (NCs). The schematic is drawn for simplicity, to illustrate the formation and rupture of conductive filaments (CFs) of Ag through the device structure.

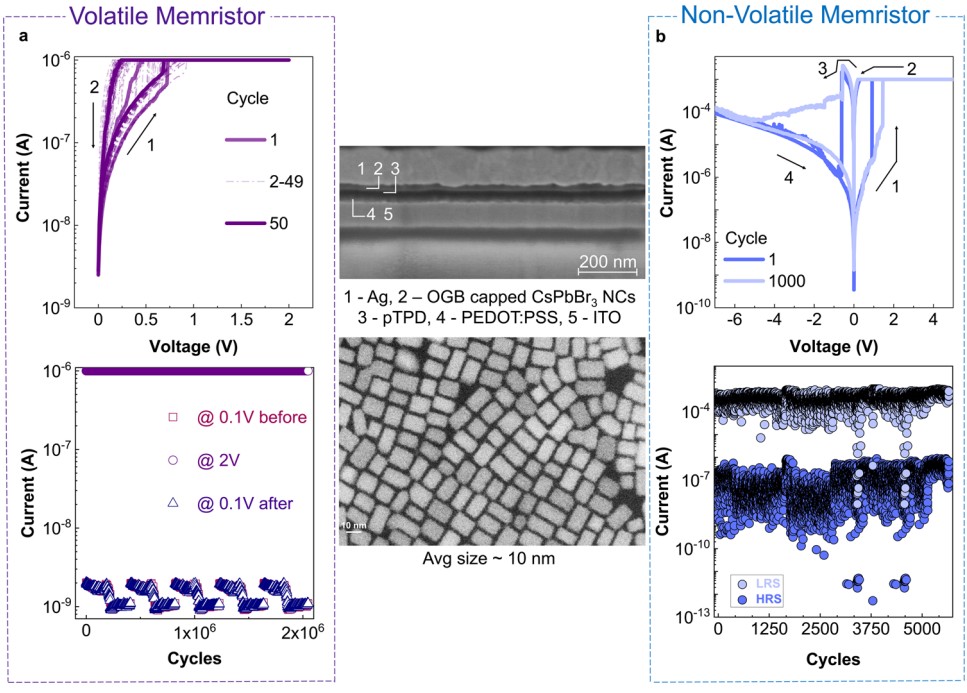

**Fig. 3 Reconfigurable halide perovskite memristor.** The device structure comprises ITO (100 nm), PEDOT:PSS (30 nm), polyTPD (20 nm), OGB-capped $CsPbBr_3$ NCs (20 nm) and Ag (150 nm) as shown in the SEM cross-section. Thickness of the individual layers were confirmed by AFM (Supplementary Note 2, Supplementary Fig. 13). The TEM image reveals NCs with an average diameter of ~10 nm. **a** Diffusive mode- evolution of the device conductance upon applying DC sweep voltages (0 V → 2 V → 0 V) with an $I_{cc} = 1 \mu A$ (top), endurance performance (bottom). **b** Drift mode- evolution of the device conductance upon applying DC sweep voltages (0 V → + 5 V → 0 V → − 7 V → 0 V) with an $I_{cc} = 1 mA$ during SET operation (top), endurance performance (bottom).

redistribution of ions, and electrochemical reactions under an applied electric field[33,34]. The larger $I_{cc}$ of 1 mA results in permanent and thicker conductive filamentary pathways, and the switching dynamics is now dominated by the drift kinetics of the mobile ion species $Ag^+$ and $Br^-$, rather than diffusion.

In the case of DDAB-capped $CsPbBr_3$ NCs, the inferior volatile endurance, quick transition to a non-volatile state and mediocre non-volatile endurance indicates poor control of the underlying electrochemical processes and formation of permanent conductive filaments even at low compliance currents. On the other

hand, capping $CsPbBr_3$ NCs with OGB ligands enables better regulation of the electrochemical processes, resulting in superior on-off ratio, volatile endurance as well as non-volatile endurance. Scanning Electron Microscope (SEM) images indicate similar film thickness in both devices, ruling out dependence on the active material thickness (Fig. 3 and Supplementary Note 2, Supplementary Fig. 9). Transmission Electron Microscopy (TEM) and Atomic Force Microscopy (AFM) images reveal similar nanocrystal size (~10 nm) and surface roughness for both films, dismissing variations in crystal size and morphology as possible

differentiating reasons (Fig. 3 and Supplementary Note 2, Supplementary Figs. 10–11). While the exact mechanism is still unknown, the larger size of the OGB ligands compared to DDAB (2.3 nm vs. 1.7 nm) could intuitively provide better isolation to the $CsPbBr_3$ NCs and prevent excess electrochemical redox reactions of $Ag^+$ and $Br^-$, modulating the formation and rupture of conductive filaments (Supplementary Note 1). This comparison is further supported by photoluminescence measurements, pointing to a larger drop of luminescence quantum yield in the films of DDAB-capped NCs, arising from the stronger excitonic diffusion and trapping (Supplementary Note 2, Supplementary Fig. 12).

To probe the mechanism further, devices with Au as the top electrode were fabricated, but did not show any resistive switching behaviour (Supplementary Note 2, Supplementary Fig. 14). The devices do not reach the compliance current of 1 mA during the set process and do not portray the sudden increase in current, typical of filamentary memristors. This indicates that Ag is crucial for resistive switching and also proves that Br- ions play a trivial role in our devices if any. Control experiments on PEDOT:PSS only and PEDOT:PSS + pTPD devices further reiterate importance of the perovskite NC thin film as an active matrix for reliable and robust Ag filament formation and rupture (Supplementary Note 2, Supplementary Figs. 15–16). Secondary ion mass spectrometry (SIMS) profiling reveals a clear difference in the $^{107}Ag$ cross section profile when comparing an ON and OFF device. An increase of the $^{107}Ag$ count is observed at the interface between the halide perovskite and the organic layers for the device in the ON state, as shown in Supplementary Note 2, Supplementary Fig. 17. Temperature-dependent measurements further confirm the theory of migration of $Ag^+$ ions through the perovskite matrix (Supplementary Note 2, Supplementary Fig. 18). The conclusions in this study are observed to be independent of the NC layer thickness, NC size and dispersity as shown in Supplementary Note 2, Supplementary Figs 19–21.

**Reservoir computing with perovskite memristors**. To demonstrate the advantages of the reconfigurability features of our perovskite memristors, we model a fully-memristive RC framework with dynamically-configured layer of virtual volatile reservoir nodes and a readout ANN layer with non-volatile weights in simulation. In particular, we address three distinct forms of computational requirements using the reconfigurability of the proposed device: an accumulating/decaying short-term memory for the temporal processing in the reservoir; a stable long-term memory for retaining trained weights in the readout layer, and a circuit methodology for accessing analog states from binary devices to enhance the training performance.

**Diffusive perovskite memristors as reservoir elements**. To implement the reservoir layer with the fabricated memristor devices, we utilize the virtual node concept originally proposed by Appeltant et al.[53]. Instead of conventional transforming of the input signal to a high-dimensional reservoir state by processing over many non-linear units, the virtual node concept employs the idea of delayed feedback on a single physical device exhibiting strong short-term effects. Under the influence of a sequential input, the dynamical device state goes through a non-linear transient response, which is recorded with fixed timesteps to create a set of virtual nodes representing the reservoir state. Hence, the transient device non-linearity constitutes temporal processing, and the delay system forms the high dimensional representation in the reservoir.

Elements of a reservoir layer should ideally possess a fading memory (sometimes called short-term memory or echo state property) and non-linear internal dynamics[54]. The fading memory effect plays a key role in extracting features in the temporal domain of the input data stream, while the non-linear internal dynamics enable projection of temporal features to a high-dimensional state with good separability[55]. Response of the OGB-capped $CsPbBr_3$ NC memristors to low-voltage electrical spikes reveal short-term/fading diffusive dynamics with a relaxation time ≥5 ms for an input pulse duration = 20 ms and amplitude = 1 V. Non-linear internal dynamics are evident in 4 formats- (i) from the transient evolution of the device conductance during the stimulations; and from the final device conductance as a function of the applied pulse (ii) amplitude, (iii) width and (iv) number (Supplementary Note 3, Supplementary Fig. 22). An additional test of the echo state property reveals that the present device state is reflective of the input temporal features in the recent past (<23 ms) but not the far past, enabling efficient capture of short-term dependencies in the input data stream (Supplementary Note 3, Supplementary Fig. 23). Stimulation of pulse streams with different temporal features results in distinct temporal dynamics of memristor states (Supplementary Note 3, Supplementary Fig. 24)

**Drift perovskite memristors as readout elements**. Storing the weight of the fully-connected readout layer of the ANN requires non-volatile synaptic devices. For representing synaptic efficacy, we use the drift-based perovskite memristor configuration that enables stable access to multiple conductance states. Because synaptic efficacy in ANNs can be either positive or negative, we use two memristor devices $G^+$ and $G^-$ in a differential architecture to represent a single synapse[56]. Hence, synaptic potentiation is obtained by increasing the conductance of $G^+$, and depression by increasing the conductance of $G^-$ with identical pulses. The effective synaptic strength is expressed by the difference between the two conductances ($G^+$–$G^-$). Arranged in a crossbar array with the differential configuration, synaptic propagation at the readout layer is realized efficiently, governed by Kirchhoff's Current Law and Ohm's Law at $O(1)$ complexity[57].

Like most filament-based memristors, our devices display non-volatile switching across only two stable states (binary) and suffer from the lack of access to true analog conductance states for synaptic efficacy. This low bit resolution during learning has been empirically shown to cause poor network performance[58,59]. To have more granular control over the filament formation, we migrate a recently proposed programming approach for oxide memristors to halide perovskites[60]. We achieve multi-level stable conductance states in the device's low resistance regime by modulating the programming $I_{cc}$. In comparison to the undesirable non-linear transformations seen in $HfO_2$ devices, the mapping from $I_{cc}$ to conductance follows a linear relation for the drift-based $CsPbBr_3$ NC devices, hence providing linear mapping to the desired conductance values (see below). This enables controlled weight updates using a single shot without requiring a write-verify scheme.

We use $I_{cc}$ modulation to train the readout layer of the reservoir network (see "Methods") using the statistical measurement data from the devices. For every input pattern received from reservoir nodes, the readout layer produces a classification prediction via a sigmoid activation function. Depending on the classification error, the desired conductance changes of each differential memristor pair per synapse are calculated. The memristive weights are then updated with the corresponding $I_{cc}$, resulting in the desired conductance values.

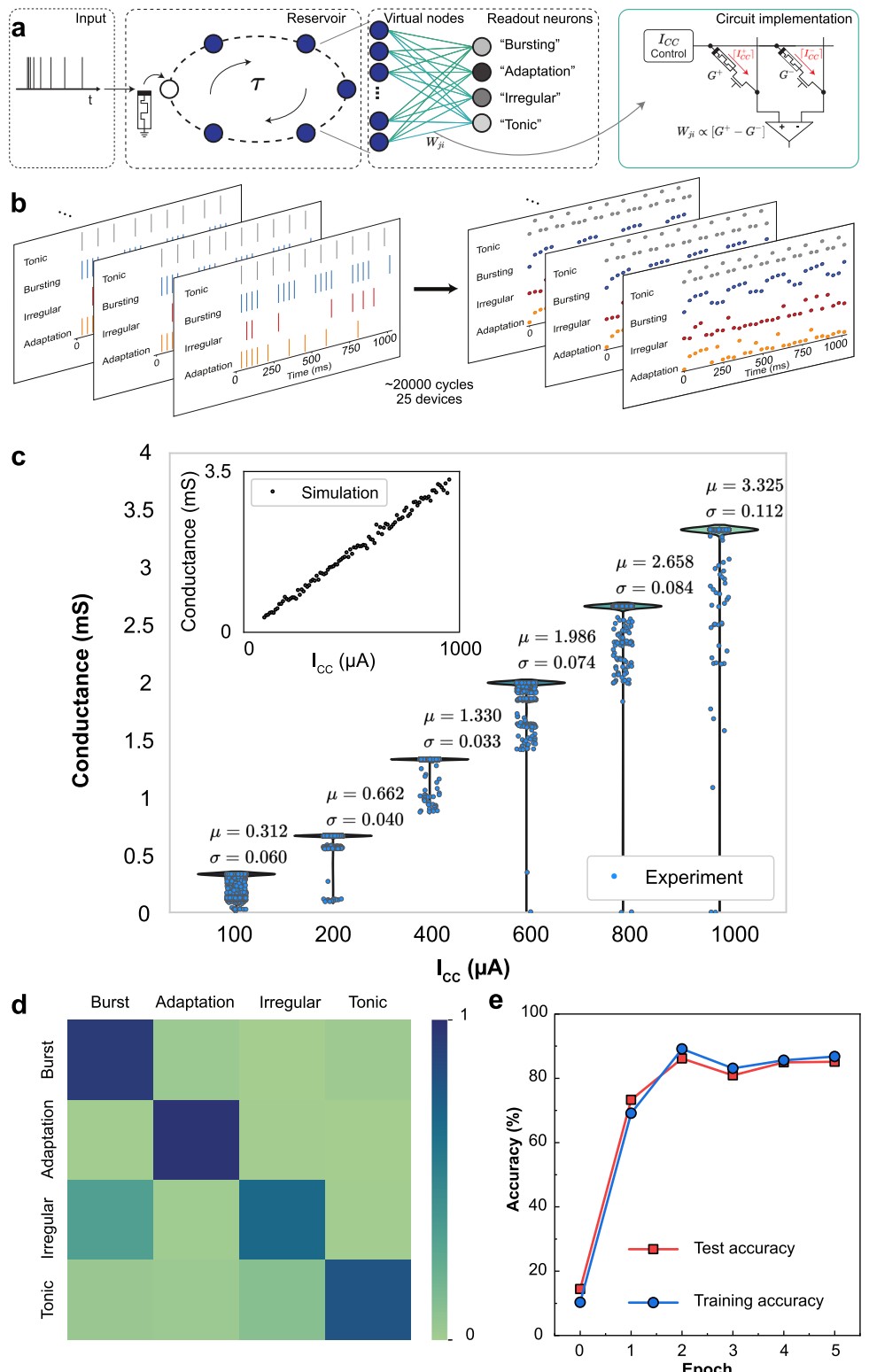

**Classification of neural firing patterns**. Next, we present a virtual reservoir neural network[53,61] simulation with the short-term diffusive configuration of perovskite memristors in the reservoir layer and long-term stable drift configuration in the trainable readout layer (Fig. 4a). The network is tested on the classification of the four commonly observed neural firing patterns in the human brain-Bursting, Adaptation, Tonic, and Irregular[62]. These spike trains (Supplementary Note 4, Supplementary Fig. 25) are applied to a single perovskite memristor in the reservoir layer, whose diffusive

dynamics constitute a short-term memory between 5 and 20 ms timescale. We exploit the concept of a virtual reservoir, where each reservoir node is uniformly sampled at finite intervals to emulate the rich non-linear temporal processing in reservoir computing. We use a sampling interval of 35 ms, resulting in a population of 30 virtual reservoir nodes representing the temporal features across 1050 ms long neural firing patterns. The device responses are derived from electrical measurements of 25 different memristive devices (Fig. 4b). Both device-to-device and

**Fig. 4 Fully-memristive reservoir computing framework with reconfigurable halide perovskite devices. a** An ANN is trained to perform classification using the temporal properties of the reservoir, in response to a series of inputs representing neural firing patterns. Using $I_{cc}$ control, OGB-capped CsPbBr$_3$ NC memristors are configured to the diffusion-based volatile mode to serve as virtual nodes in the reservoir; and to the drift-based non-volatile mode to implement synaptic weights in the ANN readout layer. During single inference, a neural firing pattern represented as a short-voltage pulse train is applied to a single diffusive-mode perovskite device. Based on the virtual node concept[53], temporal features of the input signal are intrinsically encoded as an evolving conductance of the device due to their nonlinear short-term memory effects. This evolving device state is sampled with equal intervals of 35 ms in time, denoting 30 virtual nodes that jointly represent the reservoir state. These virtual node states are delayed and fed into the readout layer, whose weights, $W_{ji}$ (size of 30 × 4), are implemented by the drift-mode non-volatile perovskite memristors, placed in a differential configuration[69]. Classification of neural firing patterns. **b** Experiments. The memristive reservoir elements are stimulated using four common neural firing input patterns - "Bursting", "Adapting", "Tonic" and "Irregular". During the presentation of inputs, the evolution of the device conductance is monitored. Each spike in the input data stream is realized as a voltage pulse of 1 V amplitude and 20 ms duration, while the device states are read with −0.5 V, 5 ms pulses. **c** Distribution of the programmed perovskite memristor non-volatile conductances with $I_{cc}$ modulation. The inset shows the simulated linear $I_{cc} \rightarrow G$ relation. Simulations. **d** Normalized confusion matrix shows the classification results with the $I_{cc}$ controlled training scheme. The RC performs slightly worse in irregular patterns due to lack of temporal correlations among samples. **e** Training (86.75%) and test (85.14%) accuracies of the fully-memristive RC framework.

cycle-to-cycle variability are captured with extensive measurements. Stimulation with "Bursting" spikes results in an accumulative behaviour within each high-frequency group and an exponential decay in the inter-group interval, reflective of the fading memory and non-linear internal dynamics as described above. "Adaptive" patterns trigger weakened accumulative behaviour as a function of the pulse interval, "Irregular" results in random accumulation and decay, while "Tonic" generates states with no observable accumulation. As the last stage of computation, these features are projected to a fully-connected readout layer with 4 sigmoid neurons (see "Methods"). The reservoir network achieves a classification accuracy of 85.1% with the training method of modulating the programming $I_{cc}$ of drift-based perovskite weights in the readout layer (Fig. 4c). Remarkably, with double-precision floating-point weights trained with the Delta rule[63] on readout, the test accuracy is 91.8% confirming the effectiveness of our $I_{cc}$ approach (Supplementary Note 4, Supplementary Figs. 26–28). The training and test accuracy over 5 epochs demonstrates that both networks are not overfitting the training data (Fig. 4d, Supplementary Note 4, Supplementary Table 1).

## Discussion

We present robust halide perovskite NC-based memristive switching elements that can be reconfigured to exhibit both volatile diffusive and non-volatile drift dynamics. This represents a significant advancement in the experimental realization of memristors. In comparison to pristine volatile and non-volatile memristors, our reconfigurable CsPbBr$_3$ NC memristors can be utilized to implement both neurons and synapses with the same material/device platform and adapt to diverse computational primitives without additional modifications to the device stack at run-time. The closest comparison to our devices are dual functional memristors- those that exhibit both volatile and non-volatile switching behaviours without additional materials or device engineering (Supplementary Note 5 Supplementary Table 2). While impressive demonstrations of dual functional memristors exist, many devices require an electroforming step to initiate the resistive switching behaviour and most importantly, the endurance and retention performance are often limited to <500 cycles in both modes and ≤10$^4$ s respectively. In comparison, we report a record-high endurance of 2 million cycles in the volatile mode, 5655 cycles in the non-volatile mode, and a retention of 10$^5$ s, highlighting the significance of our approach. This makes these devices ideal for always-on online learning systems. The forming-free operation, and low set-reset voltages would allow low power vector-matrix multiplication operations, while the high retention and endurance ensure precise mapping of synaptic weights during training and inference of artificial neural networks. In contrast to most metal oxide-based diffusive

memristors that require high programming currents to initiate filament formation (≥1 V or/and ≥10 μA), our devices demonstrate forming-free volatile switching at lower voltages and currents (≤1 V and ≤1 μA). This is possibly due to the lower activation energy for Ag$^+$ and Br$^-$ migration in halide perovskites compared to oxygen vacancies in oxide dielectrics, softer lattice of the halide perovskite layer and the large availability of mobile ion species in the halide perovskite matrix. Most importantly, our devices can be switched to the volatile mode even after programming multiple non-volatile states, proving true "reconfigurability" (Supplementary Note 5, Supplementary Fig. 29). Such behaviour is an example of the neuromorphic implementation of synapses in SNNs that demand both volatile and non-volatile switching properties, simultaneously (see Fig. 1a). It is important to note that existing implementations of dual functional devices cannot be reconfigured back to the volatile mode once the non-volatile mode is activated, making our device concept and its use case for neuromorphic computing unique.

In operando thermal camera imaging provides further support to our hypothesis of better management of the electrochemical reactions with the OGB ligands when compared to DDAB, and points to the importance of investigating nanocrystal-ligand chemistry for the development of high-performance robust memristors (Supplementary Note 6, Supplementary Figs. 30–31). While the exact memristive mechanism is still unclear, our results favour NC film implementations over thin films empirically (Supplementary Note 7, Supplementary Figs. 32–33). The insights derived on the apt choice of the capping ligands paves way for further investigations on nanocrystal-ligand chemistry for the development of high-performance robust memristors. The ability to reconfigure the switching mode on-demand allows easy implementation of multiple computational layers with a single technology, alleviating the hardware system design requirements for new neuromorphic computational frameworks. Our work complements and goes beyond previous model-based implementations[46], by comprehensively characterizing diffusive and drift devices for ~5000 patterns of different input spike streams, and collecting statistical data on device-to-device and cycle-to-cycle variability, device degradation, temporal conductance drift and real-time nanoscopic changes in memristor conductance. This statistical data is incorporated in the simulations for a very accurate modelling of the device behaviour for this task. To the best of our knowledge, this is the first time this extent of systematic analysis is being done to use the same device for both diffusive and drift behaviour for a real-world benchmark. Given the excellent performance and record endurance of our reconfigurable halide perovskite memristors, this opens way for a completely novel type of memristive substrate, for applications such as time series forecasting and feature classification.

## Methods

**Device fabrication.** Indium tin oxide (ITO, $7\,\Omega\,cm^{-2}$) coated glass substrates were cleaned by sequential sonication in helmanex soap, distilled water, acetone, and isopropanol solution. Substrates were dried and exposed to UV for 15 min. PEDOT:PSS films were deposited by spin-coating (4000 rpm for 25 s) the precursors (Clevios, Al 4083) and followed by annealing at 130 °C for 20 min. PolyTPD (Poly[N,N'-bis(4-butylphenyl)-N,N'-bisphenylbenzidine]) dissolved in chlorobenzene (4 mg/ml) was then spin-coated at 2000 rpm, 25 s; followed by annealing at 130 °C for 20 min. Solutions of $CsPbBr_3$ NCs capped with DDAB and OGB (refer to Supplementary Note 1) were next deposited via spin coating (2000 rpm for 25 s). Finally, ~150 nm of Ag was thermally evaporated through shadow masks (100 μm x 100 μm) to complete the device fabrication.

**Thin-film characterization and Electrical Measurements.** Stoe IPDS II diffractometer modified for the characterization of nano-materials (NANO-DIFF) was used to record the XRD pattern of the films. FIB-SEM (Helios 5 UX, Thermofisher Scientific) was utilised to analyse the device cross-section and JEM-2200FS JEOL TEM was used to capture images of the perovskite nanocrystals. For scanning transmission electron microscopy (STEM), a droplet of the suspension was deposited on a thin carbon foil supported on a Cu TEM grid. After drying, the specimen was mounted on a cryo holder (Gatan 626). After cooling down to liquid nitrogen temperature and drift stabilization of the specimen, images were recorded with a high-angle annular dark field detector (HAADF-STEM) on an aberration-corrected dedicated STEM microscope, a Hitachi HD-2700Cs (frame time 5–20 s). Absorption and steady-state PL spectra were collected using Jasco V 770 and FluoroMax FL1013, respectively. Electrical measurements were carried out using a Karl Suss PM8 Manual Probe Station and Keithley 4200 SCS under ambient conditions without any encapsulation.

Note: For endurance testing in the volatile mode, write and read voltages of + 2 V and + 0.1 V were used respectively with a pulse width of 5 ms. The following methodology was used: 1. read the current level of the device using + 0.1 V, 2. apply + 2 V for 5 ms as the write pulse and monitor the device's current level, 3. repeat step 1. For the non-volatile mode, write voltage of + 5 V, erase voltage of − 7 V and read voltage of + 0.1 V were used. The following methodology was used: 1. read the current level of the device using + 0.1 V, 2. apply + 5 V/−7 V for 5 ms as the write/erase pulse, 3. repeat step 1 and extract the on-off ratio comparing steps 1 and 3. Note: Since our VM loses the stored information upon removing power, the ON state ($I_{power\ ON}$) is reported as the current value corresponding to the application of the programming pulse (at 2 V) and the OFF state ($I_{power\ OFF}$) is reported as the current value corresponding to the application of the reading pulse (at 0.1 V), in alignment with the reported literature[10]. For endurance measurements in the non-volatile memory (NVM) mode, the conventional methodology was used, i.e., the ON-OFF ratios were extracted from the current values corresponding to the same reading pulse (0.1 V).

**Neural spike pattern dataset generation.** The neural spike pattern dataset consists of samples of four classes: Bursting, Adaptation, Tonic, and Irregular. "Bursting" firing patterns are defined as groups of high-frequency spikes with a constant inter-group interval; "Adapting" corresponds to spikes with gradually increased intervals; "Tonic" denotes low-frequency spikes with a constant interval; and "Irregular" corresponds to spikes that fire irregularly. In total, the dataset consists of 4975 patterns (199 cycles applied to 25 devices) for each of the four types. Each pattern is ~1050 ms long, where spikes are emulated with square wave voltage pulses (1 V, 25 ms). For Bursting patterns, each spike train consists of 4–5 high-frequency burst groups (4 spikes per burst group) with an interspike interval (ISI) of 5 ms. Between bursts, there exist 75–125 ms intervals. For Adaptation patterns, each spike train starts with high-frequency pulses with an ISI of 5 ms and gradually increases 50% with each new spike (with 5% standard deviation). For Tonic patterns, a regular spiking pattern with an average ISI of 70 ms is used. For each ISI, 5% standard deviation is applied. For irregular patterns, spike trains are divided into 60 ms segments, and a spike is assigned randomly with a 50% probability to the beginning of each segment.

**Simulation of neural networks.** For classifying neural spike patterns, a fully-connected readout layer with 30 inputs and 4 outputs is used. In addition, there is one bias unit in the input. The 4 neurons at the output are sigmoid neurons. For training, 90% of the neural spike pattern dataset is used over 5 epochs. At the end of each epoch, the network performance is tested with the rest 10% of the dataset. During $I_{cc}$ modulated training, each synapse comprises two conductance values in a differential configuration. The differential current is scaled such that $W = \beta\ (G^+ − G^-)$, where $\beta = 1/(G_{max} − G_{min})$, corresponds to maximum and minimum allowed conductance values of memristors. Conductances are initialized randomly with a Normal distribution ($\mu_G = 0.5$ mS and $\sigma_G = 0.1$ mS). Network prediction is selected deterministically by choosing the output neuron with the maximum activation. After the prediction, the L1 loss is calculated. Then, weight change that reflects a step in the direction of the ascending loss gradient is calculated with $\Delta W = (\eta\ x_i\ \delta_j)/\beta$, where $\eta$ is the learning rate, $x_i$ is the reservoir node output, $\delta_j$ is the calculated loss and $1/\beta$ is the scaling factor between weights and conductances. Target weights are clipped between 0.1 mS and 3.5 mS. Subsequently, $I_{cc}$ values corresponding to the target conductances are calculated (see Supplementary Note 4, Supplementary Fig. 28). Finally, we sample new conductance values from a Normal distribution whose mean and standard deviation is calculated using linear functions of $I_{cc}$. For the double-precision floating-point-based training, the same readout layer size is used. Network loss is calculated via the Mean Squared Error. Weights are adjusted using the Delta rule with an adaptive learning rate[64]. Both networks are trained with a batch size of 1 and a suitably tuned hyperparameters.

## Data availability

The authors declare that the main data supporting the findings of this study are available within the paper and its Supplementary Information files. Source data are provided with this paper.

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

## Acknowledgements

R.A.J. acknowledges the support from the ETH Zurich Postdoctoral Fellowship scheme for this work. Y.D. acknowledges the funding from European Union's H2020 research and innovation programme under the Marie Skłodowska-Curie grant agreement No 861153. P.Z., F.K. and M.I.B. acknowledge the support of the Scientific Center for Optical and Electron Microscopy (ScopeM) of the Swiss Federal Institute of Technology (ETHZ) and Empa Electron Microscopy Center. R.A.J. acknowledges the help of Roman Furrer (Empa) for the thermal camera imaging experiments.

## Author contributions

R.A.J. conceived the project direction and experiments. Y.D., M.P. and M.V.N. formulated the RC framework simulations. Y.B. synthesized the perovskite nanocrystals and performed the basic physical characterizations under the supervision of M.I.B. and M.V.K. R.A.J. and Y.S. fabricated the devices. R.A.J. performed all the electrical characterizations and AFM imaging with the help of G.K. and I.S. Y.D. and M.P. carried out the neural network simulations under the supervision of G.I. N.O. performed the SIMS experiments and analysis with the help of G.A.C. and guidance from T.L.. P.Z. and F.K. performed the FIB cut, SEM and STEM imaging. M.I.B. performed TEM imaging of the nanocrystals. R.A.J. and Y.D. wrote the manuscript with inputs from all authors. R.A.J. and Y.D. contributed equally to this work. G.I and M.K. supervised the project overall.

## Competing interests

The authors declare no competing interests.
