## [Peer Review File · Nature Communications]

Reconfigurable Halide Perovskite Nanocrystal Memristors for Neuromorphic ComputingREVIEWER COMMENTS

Reviewer #1 (Remarks to the Author):

John et al. report a study on reconfigurable halide perovskite nanocrystal memristors for neuromorphic computing. The primary stated fundamental advance in the study is the ability to reconfigure devices between volatile (diffusive) and non-volatile (drift) modes by controlling electrochemical reactions. The primary technological/applied advance is benchmarking 25 devices across numerous measurements to ultimately demonstrate high endurance in both the volatile and non-volatile modes. I think that the authors succeed on the technological side by demonstrating the application durable and reconfigurable devices and an interesting reservoir network with recurrent and read-out layers. However, I feel that the fundamental advance is not clearly demonstrated, as the authors primarily make many blanket mechanistic statements about how the devices perform without the appropriate control samples, control experiments, and ultimate data to back up these statements. I'm left wondering how to apply the findings to more generalizable design principles for these types of reconfigurable devices, but I don't think the authors have provided enough insights to make such conclusions possible. Some pertinent questions, elucidated in more detail below, include the roles of: NC surface sites/chemistry, ligand identity and coverage, NC layer thickness, NC size, halide versus electrode ion diffusion/migration, contact layers, nanocrystalline versus bulk perovskite performance. As such, I would not recommend the publication of the manuscript in its current state. It would require a major revision, incorporating key systematic control samples and experiments, for the authors to convincingly demonstrate that they are "controlling electrochemical reactions" in a way that the field can take and reproduce.

- On page 4, the authors state "While most reports are based on thin films or bulk crystals of halide perovskites, interestingly perovskite nanocrystal-based formulations^{37,38} have been overlooked till date, entailing significant research in this direction." Here, the authors cite a couple of their own (general, not neuromorphic or memory-focused) review articles but ignore significant prior literature in the field. While perovskite nanocrystal-based memory elements may be less prevalent than films/crystals in the literature thus far, it is inaccurate to say that they have not been studied and reported in the literature for these types of applications. Here are several examples:

- Appl. Phys. Lett. 2017, 110, 083102.
- Adv. Mater. 2018, 30, 1800327.
- Adv. Mater. 2018, 30, 1802883.
- ACS Appl. Mater. Interfaces 2019, 11, 24367.
- Sci. Adv. 2021, 7, eabf1959.

- There are many mechanistic statements that are made in the manuscript without data to back them up (see bulleted list below). Do the authors have data that demonstrate e.g. filament formation, Ag and/or Br movement, etc. under different conditions or are these statements just based on generalized expectations? If the latter, it isn't clear if the authors are basing these statements on the general expectations for e.g. a bulk perovskite system of similar composition, or if they're basing the statements on the general expectations of this type of system (i.e. expected diffusion/migration properties and filament formation properties within perovskite NC layers). The omission of any perovskite NC studies from the citation list (see above) makes me think it is the former (expectations for bulk perovskites). Regardless, several of the NC studies listed above (and prior studies on bulk perovskites) use spatially and chemically sensitive techniques (SEM, EDX, SIMS, etc.) to track these types of processes. The lack of such data in the current study makes it hard to accept the mechanisms proposed by the authors and also makes it difficult to understand how these results may be applicable to more generalizable design principles for similar devices (see also questions below regarding contributions of surface versus bulk, etc.).

- o "In their diffusive mode, the low activation energy of ion migration of the mobile ionic species (Ag⁺ and Br⁻) enables volatile switching."

- o "In the drift mode, stable conductive filaments formed by the drift of the ionic species facilitate programming of non-volatile synaptic weights in the readout layer..."

- o "The volatile threshold switching behaviour can be attributed to the redistribution of Ag⁺ and Br⁻ ions under an applied electric field, and their back-diffusion upon removing power (Fig. 2, Supplementary Note 2, Supplementary Figure 3)^{34,46,47}."

- o "In the case of DDAB-capped CsPbBr₃ NCs, the inferior volatile endurance, quick transition to a non-volatile state and mediocre non-volatile endurance indicates poor control of the underlying electrochemical processes and formation of permanent conductive filaments even at low compliance currents."

- o "While the exact mechanism is still unknown, the larger size of the OGB ligands compared to DDAB (2.3 nm vs. 1.7 nm) could intuitively provide better isolation to the CsPbBr₃ NCs and prevent excess electrochemical redox reactions of Ag⁺ and Br⁻, modulating the formation and rupture of conductive filaments."

- How thick is the perovskite nanocrystal thickness in these devices? The only way I can gauge this is to look at the 200 nm scale bar of Figure 2's SEM cross section. From that scale bar, it looks like the layer (layer 2 in the SEM image) could be 10 – 20 nm thick, which would essentially make it one monolayer of nanocrystals. Is this what the authors were aiming for and expecting? Do the device characteristics depend sensitively on this nanocrystal layer thickness? Do the device characteristics depend on the average nanocrystal size or polydispersity?

- Based on the question above and the SEM cross section generally, I'm very confused by the schematics in Figure 2. The pTPD layer looks to be significantly thicker than both the NC layer and the PEDOT:PSS layer in the SEM, but the thickest layer in the schematics is the NC layer. The NC layer is also depicted as a continuous slab, almost like a bulk semiconductor. But this is not the case, as it is somewhere between 1 and X layers thick of a NC array and there are bulk sites and (many) surface sites. The schematics bring up more questions than they answer. Some relevant questions:

- o Where is the ion diffusion and migration occurring? On the surface of the NCs? Within the bulk of the NCs? If surfaces are important, what is the role being played mechanistically by the ligand?

- o Have the authors considered temperature-dependent measurements to confirm that the activation energies are consistent with ion movement?

- o Does it actually matter that these are CsPbBr₃ NCs or would the same effects occur for a CsPbBr₃ thin film? At such a thin layer (and given the level of detail of schematics in Figure 2), it would seem like a CsPbBr₃ thin film may work just as well.

- o Why are pTPD and PEDOT:PSS chosen specifically? What role do they play in ion migration, charge transport, etc.?

- o Does the silver have to be in direct contact with the perovskite NC layer for the devices to function? Do they function similar if a transport layer is inserted between the Ag electrode and the perovskite layer and can this help inform about the role of filament formation?

Reviewer #2 (Remarks to the Author):

The authors have demonstrated a reconfigurable halide perovskite nanocrystal memristor that achieves on-demand switching between diffusive/volatile and drift/non-volatile modes. Moreover, the CsPbBr₃ nanocrystals capped with oleylguanidinium bromide (OGB) ligands exhibit remarkable endurance performances in both volatile and non-volatile modes. In addition, the authors constructed a fully-memristive reservoir computing framework with diffusive perovskite memristors as reservoir elements and drift perovskite memristors as readout elements. However, the following issues should be addressed.

1. The authors defined that the diffusive mode is volatile and the drift mode is nonvolatile. Please clearly tell the criteria to distinguish the volatile mode from the nonvolatile mode?

2. In Figure 2, the endurance measurements show an inferior non-volatile endurance than the volatile endurance. Please discuss the underlying mechanisms.

3. The I-V curve in Figure 2a only shows the positive voltage range. Does the I-V curve remain the same in the reverse voltage sweep?

4. In Figure 3b, there are 25 devices in the simulation of the reservoir computing. Should the uniformity of the 25 devices affect the accuracy of the training and test process? Does the uniformity need to be considered.

5. The second paragraph is not well logically connected to the preceding text. Please improve the writing.

6. Are both the diffusive and drift mechanisms essentially related to the movement of ions and vacancies under an electric field? The difference lies in the fact that one forms a conducting filament, which can remain in a low resistance state for a long time. This is the so-called non-volatile state. But the other does not form a conducting filament. The low resistance state will revert to the high resistance state in a short time. This is the so-called volatile state. Please make appropriate clarification in the manuscript.

7. Please explain why the formation of the conductive filament is forming-free? How does the ICC exactly regulate the threshold switching?

8. Why are the current organic ligands chosen? What are the molecular structures of the two organic ligands? Why do the perovskite nanocrystals with the DAAB ligands exhibit far inferior properties compared to the perovskite nanocrystals of the OGB ligands? Is it because the length of the molecular chain does not have the same segregation effect on Ag^+ and Br^- ? The results of the PL measurements do not seem to be sufficient evidence for the vast difference in performance between the two ligands.

9. What is the delayed system? Please give the definition of separability in this system? How does this relate to the characteristics of the device?

10. Why is the conducting filament described by using the word "thick"? Why can a thick filament not be erased by negative bias?

Reviewer #1 (Remarks to the Author):

John et al. report a study on reconfigurable halide perovskite nanocrystal memristors for neuromorphic computing. The primary stated fundamental advance in the study is the ability to reconfigure devices between volatile (diffusive) and non-volatile (drift) modes by controlling electrochemical reactions. The primary technological/applied advance is benchmarking 25 devices across numerous measurements to ultimately demonstrate high endurance in both the volatile and non-volatile modes. I think that the authors succeed on the technological side by demonstrating the application durable and reconfigurable devices and an interesting reservoir network with recurrent and read-out layers. However, I feel that the fundamental advance is not clearly demonstrated, as the authors primarily make many blanket mechanistic statements about how the devices perform without the appropriate control samples, control experiments, and ultimate data to back up these statements. I'm left wondering how to apply the findings to more generalizable design principles for these types of reconfigurable devices, but I don't think the authors have provided enough insights to make such conclusions possible. Some pertinent questions, elucidated in more detail below, include the roles of: NC surface sites/chemistry, ligand identity and coverage, NC layer thickness, NC size, halide versus electrode ion diffusion/migration, contact layers, nanocrystalline versus bulk perovskite performance. As such, I would not recommend the publication of the manuscript in its current state. It would require a major revision, incorporating key systematic control samples and experiments, for the authors to convincingly demonstrate that they are "controlling electrochemical reactions" in a way that the field can take and reproduce.

We thank the Referee for the positive comments on our work and for acknowledging the merit of our technological advance in terms of device properties and reservoir computing demonstration. We also very much appreciate the Reviewer's critical suggestions on mechanistic understanding of our devices to improve the manuscript. Our responses to the comments are as follows.

• On page 4, the authors state "While most reports are based on thin films or bulk crystals of halide perovskites, interestingly perovskite nanocrystal-based formulations^{37,38} have been overlooked till date, entailing significant research in this direction." Here, the authors cite a couple of their own (general, not neuromorphic or memory-focused) review articles but ignore significant prior literature in the field. While perovskite nanocrystal-based memory elements may be less prevalent than films/crystals in the literature thus far, it is inaccurate to say that they have not been studied and reported in the literature for these types of applications. Here are several examples:

- Appl. Phys. Lett. 2017, 110, 083102.
- Adv. Mater. 2018, 30, 1800327.
- Adv. Mater. 2018, 30, 1802883.
- ACS Appl. Mater. Interfaces 2019, 11, 24367.
- Sci. Adv. 2021, 7, eabf1959.

We thank the Reviewer for the comment. We would like to clarify that our intention was not to exclude halide perovskite nanocrystal (NC)-based studies in literature, but was to convey that such NC formulations are much less investigated in comparison to thin films. We apologize for not having cited more relevant references. Moreover, we did not consider synaptic transistor implementations (most of the above mentioned manuscripts) as direct comparison against 2-terminal conventional memristor implementations because of their very different working mechanisms. However, based on the Reviewer comment, we have now amended the sentence to

"While most reports are based on thin films or bulk crystals of halide perovskites, interestingly perovskite nanocrystal (NC)-based formulations have been much less investigated till date. Existing implementations often utilize NCs only as a charge trapping medium to modulate the resistance state of another semiconductor, in flash-like configurations a.k.a synaptic transistor. The memristive switching capabilities and limits of the perovskite NC active matrix remains unaddressed, entailing significant research in this direction."

We have inserted all the references mentioned by the Reviewer and also added few others. Please refer to page 4 in the main text.

• There are many mechanistic statements that are made in the manuscript without data to back them up (see bulleted list below). Do the authors have data that demonstrate e.g. filament formation, Ag and/or Br movement, etc. under different conditions or are these statements just based on generalized expectations? If the latter, it isn't clear if the authors are basing these statements on the general expectations for e.g. a bulk perovskite system of similar composition, or if they're basing the statements on the general expectations of this type of system (i.e. expected diffusion/migration properties and filament formation properties within perovskite NC layers). The omission of any perovskite NC studies from the citation list (see above) makes me think it is the former (expectations for bulk perovskites). Regardless, several of the NC studies listed above (and prior studies on bulk perovskites) use spatially and chemically sensitive techniques (SEM, EDX, SIMS, etc.) to track these types of processes. The lack of such data in the current study makes it hard to accept the mechanisms proposed by the authors and also makes it difficult to understand how these results may be applicable to more generalizable design principles for similar devices (see also questions below regarding contributions of surface versus bulk, etc.).

- "In their diffusive mode, the low activation energy of ion migration of the mobile ionic species (Ag⁺ and Br⁻) enables volatile switching."

- "In the drift mode, stable conductive filaments formed by the drift of the ionic species facilitate programming of non-volatile synaptic weights in the readout layer..."

- "The volatile threshold switching behaviour can be attributed to the redistribution of Ag⁺ and Br⁻ ions under an applied electric field, and their back-diffusion upon removing power (Fig. 2, Supplementary Note 2, Supplementary Figure 3)^{34,46,47}."

- "In the case of DDAB-capped CsPbBr₃ NCs, the inferior volatile endurance, quick transition to a non-volatile state and mediocre non-volatile endurance indicates poor control of the underlying electrochemical processes and formation of permanent conductive filaments even at low compliance currents."

- "While the exact mechanism is still unknown, the larger size of the OGB ligands compared to DDAB (2.3 nm vs. 1.7 nm) could intuitively provide better isolation to the CsPbBr₃ NCs and prevent excess electrochemical redox reactions of Ag⁺ and Br⁻, modulating the formation and rupture of conductive filaments."

We thank the Reviewer for the comment. Based on the Reviewer's suggestion, we have now conducted several additional experiments to verify the resistive switching mechanism in our devices. Investigations with Au as the active electrode and Secondary-ion mass spectrometry (SIMS) results are presented here. Some of the other mechanistic investigations are answered below separately in response to specific questions.

1. Investigation with Au as the active electrode:

Devices with Au as the top electrode were fabricated, but did not show any resistive switching behaviour (Figure R1.1). The devices did not reach the compliance current of 1 mA during the set process and did not portray the sudden increase in current, typical of filamentary memristors. This indicates that Ag is crucial for resistive switching in our devices. This also proves that Br⁻ ions play a trivial role in our devices if any. We have now inserted this figure to the Supporting Information and discussed in the main text (page 9). Please refer to page 12 Supplementary Note 2 Supplementary Fig. 14.

Figure R1.1. Effect of Au on the switching characteristics of OGB-capped perovskite NC memristors.

2. Secondary-ion mass spectrometry (SIMS) studies:

As per the Reviewer's suggestion, we have carried out SIMS profiling on our devices. The SIMS scans reveal a clear difference in the ^{107}Ag cross section profile when comparing an ON and OFF device. In both cases, the depth profiles highlight a severe intermixing of ^{107}Ag induced by the sputtering O_2^+ ion beam. The origin of the intermixing is most likely ballistic [1] and electrically driven [2].

When isolating the effect of the intermixing by normalizing the ^{107}Ag with respect to a referenced OFF device, the ^{107}Ag profile of the ON device displays a considerable difference (Figure R1.2). The Ag pads deposited on top of the surface seem to be Ag depleted (i.e., a lower amount of Ag), which is confirmed by microscope observation. It must be noted that the first few nanometers of the depth profiling are affected by surface contaminations and can be considered as a transient region. The interesting section is located at the interface between the halide perovskite and the organic layers (near the top red line shown in the plots, also denoted by the circle), where an increase of the ^{107}Ag count is observed.

Although, the intermixing mentioned above still affects the ^{107}Ag profile, it is clearly seen by the depth profiles, that there is ^{107}Ag present in the structure due to the operation of the device. Although the ^{107}Ag cannot be presently localized precisely in the structure, a qualitative comparison of the Ag profiles between the ON and OFF states proves formation of Ag conductive filaments upon biasing, validating our proposed memristive switching mechanism.

Figure R1.2 Secondary-ion mass spectrometry (SIMS) studies. **a** Ratio of an OFF and ON device with respect to the average ^{107}Ag counts per cycle normalized to a referenced OFF device, i.e., OFF/ON. 2D representation of the ^{107}Ag located in the cross section of an **b** ON and **c** OFF device.

We have now inserted this to the Supporting Information. Please refer to pages 14-15 Supplementary Note 2, Supplementary Fig. 17 and page 9 in the main text.

References:

1. Hofmann, S. "Sputter depth profile analysis of interfaces." **Reports on Progress in Physics** 61.7 (1998): 827.
2. Vriezema, C. J., and P. C. Zalm. "Impurity migration during SIMS depth profiling." **Surface and Interface Analysis** 17.12 (1991): 875-887.

• How thick is the perovskite nanocrystal thickness in these devices? The only way I can gauge this is to look at the 200 nm scale bar of Figure 2's SEM cross section. From that scale bar, it looks like the layer (layer 2 in the SEM image) could be 10 – 20 nm thick, which would essentially make it one monolayer of nanocrystals. Is this what the authors were aiming for and expecting? Do the device characteristics depend sensitively on this nanocrystal layer thickness? Do the device characteristics depend on the average nanocrystal size or polydispersity?

Yes the Reviewer is correct that the CsPbBr₃ NC film is about 20nm in thickness. We had mentioned this explicitly in the caption of Figure 2. We have verified the thickness of individual layers via both cross-sectional scanning electron microscopy (SEM) (Figure 2) and Atomic force microscopy (AFM) (Figure R1.3).

Figure R1.3 Thickness determination. AFM scan area of PEDOT:PSS + pTPD + CsPbBr₃ NCs (left). Step height profile (right) of the white line shown on the left. Thickness of the CsPbBr₃ NC thin film is calculated in a subtractive manner as indicated by the text in the figure. **a** and **b** shows the result of DDAB and OGB-capped CsPbBr₃ NCs respectively.

In our experiments, we do not observe any direct dependence of the device characteristics on the nanocrystal layer thickness. Figure R1.4 shows the device characteristics as a function of the nanocrystal layer thickness. All devices exhibit very similar characteristics with an on-off ratio $\geq 10^3$, similar set and reset voltages. Statistical analysis of the on-off ratios also reveal independence from the nanocrystal layer thickness.

Figure R1.4 Effect of NC layer thickness. IV characteristics (top) and statistical analysis of the distribution of on-off ratios during endurance testing (bottom) of 20nm and 90nm OGB-capped CsPbBr₃ NC memristors.

To study the effects of NC size and dispersity, we prepared 3 solutions of OGB-capped NCs with an average size of 13, 9 and 7nm. Since NCs have a slightly elongated shape in one direction, it is more convenient to refer to the aspect ratio of the NCs. The corresponding aspect ratios are: 1.47 ± 0.23 , 1.26 ± 0.36 and 1.57 ± 0.25 respectively. The DDAB-capped NCs had an average size of 9nm with an aspect ratio of 1.09 ± 0.1 . The size, aspect ratio and monodispersity is confirmed via TEM (not shown here) and photoluminescence measurements (Figure R1.5). For all colloids, we observed a narrow FWHM of PL spectra (~ 18 nm for all OGB samples and ~ 20 nm for DDAB samples) that confirms their high monodispersity.

Figure R1.5. Photoluminescence spectroscopy of a DDAB and b-d OGB-capped CsPbBr₃ NCs of various sizes (13, 9 and 7nm respectively) and aspect ratios in solution.

Devices fabricated with each of the solutions for comparison exhibit very similar characteristics with an on-off ratio $\geq 10^3$, similar set and reset voltages. Statistics derived from extensive measurements also do not show any trend with the nanocrystal size (Figure R1.6). While the exact mechanism is still a matter of ongoing research, the best results in terms of yield of devices with an on-off ratio $\geq 10^3$ and endurance of > 4000 cycles in the non-volatile mode is obtained with NCs of size 9nm with an aspect ratio of 1.26 ± 0.36 . We have now inserted all these details to the Supporting Information. Please refer to page 12 Supplementary Fig. 13, pages 16-18 Supplementary Figs. 19-21 and page 10 in the main text.

Figure R1.6 Effect of NC size. IV characteristics (top) and a representative statistical analysis of the distribution of on-off ratios during endurance testing (bottom) of **a** 13nm, **b** 9nm and **c** 7nm OGB-capped CsPbBr₃ NC memristors.

• Based on the question above and the SEM cross section generally, I'm very confused by the schematics in Figure 2. The pTPD layer looks to be significantly thicker than both the NC layer and the PEDOT:PSS layer in the SEM, but the thickest layer in the schematics is the NC layer. The NC layer is also depicted as a continuous slab, almost like a bulk semiconductor. But this is not the case, as it is somewhere between 1 and X layers thick of a NC array and there are bulk sites and (many) surface sites. The schematics bring up more questions than they answer. Some relevant questions:

We would like to first clarify the thicknesses of each layer in our device: ITO -100nm, PEDOT:PSS- 30nm, pTPD- 20nm, CsPbBr₃ NC film - 20nm, Ag- 150nm. This was mentioned as the first line of Figure 2's caption on page 8. Thus, the thickness of the PEDOT:PSS and pTPD interlayers are comparable and not much larger than the perovskite NC thin film. We have systematically carried out experiments to rule out the possibility of the contribution of PEDOT:PSS and pTPD to the switching process. Please refer to the answer below.

With regards to the schematic, firstly, we would like to apologize for any misunderstanding caused. We agree with the Reviewer that the current representation in the form of a continuous slab is not the best way. However, our intention with the figure is to illustrate and highlight the formation of conductive filaments (CFs) of Ag through the device structure. We believe that this message will be best perceived to a reader by just focussing on the ions that migrate under bias. The thicknesses are not drawn to scale (to match the experimentally-measured thicknesses) to again highlight the CF formation and rupture within the perovskite layer. However, we understand the Reviewer's concern. As per the Reviewer's suggestion, we have now clarified the above in the caption of Figure 2. Please refer to the new caption of Figure 2 on page 9 of main text, also furnished below.

“Additional note: The thickness of the individual layers in the device schematic are not drawn to scale to match the experimentally-measured thicknesses. The perovskite layer is not a bulk semiconductor, but 1-2 layers of nanocrystals (NCs). The schematic is drawn for simplicity, to illustrate the formation and rupture of conductive filaments (CFs) of Ag through the device structure.”

Moreover, we will consult with the editorial office to see if a better schematic can be redrawn to illustrate the same without compromising on the structural integrity of the NC layer. We once again apologize for not having mentioned this explicitly before.

o Where is the ion diffusion and migration occurring? On the surface of the NCs? Within the bulk of the NCs? If surfaces are important, what is the role being played mechanistically by the ligand?

The dynamic ligand binding in halide perovskite (HP) nanocrystals (NCs) have been widely observed to create surface traps [1]. Therefore, handling the surface ligands without sacrificing the optoelectronic properties or affecting the structural integrity poses a challenge to developing semiconductive HP NC films. The large PL quenching observed with DDAB ligands (solution vs thin film comparison shown in Supplementary Fig. 12) point to creation of large number of surface traps. In perovskite films and NCs, it has been firmly established that trap states at the grain boundaries and on the surface can capture photo-excited charge carriers to create a local electric field capable of promoting ion migration [2-7]. In our case, this could lead to enhanced Ag^+ and Br^- migration and eventually lead to thicker Ag filament formation. From our IV endurance measurements, the DDAB devices are observed to

(i) quickly transit from a volatile to a non-volatile state, even at a low compliance current (I_{cc}) of 1 μA resulting in an inferior volatile endurance of ~ 10 cycles, and

(ii) quickly transit to a non-erasable non-volatile state at high I_{cc} of 1 mA, resulting in an inferior non-volatile endurance of ~ 50 cycles.

Both these results support the hypothesis of enhanced electrochemical reactions in the DDAB system due to their short chains.

To investigate this further, we monitored the working of DDAB and OGB memristors *in operando* using a thermal camera. We observe that maximum heat is generated in our memristors during the reset process when trying to break the conductive filament(s). Application of reverse bias causes Joule heating which ruptures the conductive filament(s). Thick filaments are difficult to break and result in large rise in temperature in the surrounding areas which we pick up as an infrared image [8-10]. Figure R1.7 shows the thermograms recorded at the time of failure of the memristors- 50 cycles for DDAB and 5655 cycles for OGB. For the OGB device, a thermogram is also recorded at the 50th endurance cycle for direct comparison to the DDAB device. At the time of failure of the DDAB device, i.e. 50th endurance cycle, a 95°C rise in surface temperature is seen. In comparison, only a 13°C rise is observed for OGB devices at the 50th cycle. When the OGB device reaches its maximum endurance/near failure (5655 cycles), a 86°C rise in surface temperature is noted similar to the DDAB device. These findings indicate that an applied field alone cannot rupture the conductive filaments and that the reset process is likely to be a combined effect of electric field and Joule heating. Hence it is critical to engineer materials to regulate the underlying electrochemical reactions. This experiment further supports our hypothesis of better management of the electrochemical reactions with the larger OGB ligands when compared to DDAB, and points to the importance of investigating nanocrystal-ligand chemistry for the development of high-performance robust memristors.

We have now inserted this discussion to the Supporting Information. Please refer to pages 29-30 Supplementary Note 6 Supplementary Fig. 31 and page 15 main text.

Figure R1.7 Thermal imaging. *In operando* monitoring of the DDAB and OGB memristors using an infrared camera.

References:

1. Xue, Jingjing, Rui Wang, and Yang Yang. "The surface of halide perovskites from nano to bulk." **Nature Reviews Materials** 5.11 (2020): 809-827.
2. DeQuilettes, Dane W., et al. "Photo-induced halide redistribution in organic–inorganic perovskite films." **Nature Communications** 7.1 (2016): 1-9.

- Azpiroz, Jon M., et al. "Defect migration in methylammonium lead iodide and its role in perovskite solar cell operation." **Energy & Environmental Science** 8.7 (2015): 2118-2127.
- Ahn, Namyoung, et al. "Trapped charge-driven degradation of perovskite solar cells." **Nature Communications** 7.1 (2016): 1-9.
- Kim, Young-Hoon, et al. "Charge carrier recombination and ion migration in metal-halide perovskite nanoparticle films for efficient light-emitting diodes." **Nano Energy** 52 (2018): 329-335.
- Kamat, Prashant V., and Masaru Kuno. "Halide ion migration in perovskite nanocrystals and nanostructures." **Accounts of Chemical Research** 54.3 (2021): 520-531.
- Chen, Mingming, et al. "Manipulating ion migration for highly stable light-emitting diodes with single-crystalline organometal halide perovskite microplatelets." **ACS Nano** 11.6 (2017): 6312-6318.
- Gogoi, Himangshu Jyoti, and Arun Tej Mallajosyula. "Enhancing the Switching Performance of CH₃NH₃PbI₃ Memristors by the Control of Size and Characterization Parameters." **Advanced Electronic Materials** (2021): 2100472.
- Li, Yibo, et al. "Review of memristor devices in neuromorphic computing: materials sciences and device challenges." **Journal of Physics D: Applied Physics** 51.50 (2018): 503002.
- Yoon, Kyung Jean, et al. "Electrically-generated memristor based on inkjet printed silver nanoparticles." **Nanoscale Advances** 1.8 (2019): 2990-2998.

o Have the authors considered temperature-dependent measurements to confirm that the activation energies are consistent with ion movement?

We begin by replotting the I–V curves on a double-log scale. The devices exhibit an Ohmic contact behaviour with a linear slope of 1 in the LRS (Figure R1.8a), supporting the hypothesized mechanism of Ag CF formation and rupture [1]. In the HRS, multiple slope features corresponding to various charge transport behaviour ($I \propto V^n$) are observed across all compositions (Figure R1.8b), pointing to trap-controlled space-charge-limited currents (SCLC) due to trapping and detrapping of injected electrons by the inherent defects in the perovskite matrix [2]. The exact dynamics in the HRS is still unclear due to hidden contributions of halide ion migration in perovskites [3-4]. To calculate the activation energy, we investigate the variation in the HRS in the voltage range 0.01-0.1V as a function of temperature using the Arrhenius equation $R = R_0 e^{-E_a/k_B T}$ (Figure R1.8c). The extracted value of 0.208eV matches values reported in literature for Ag⁺ migration [5], supporting the hypothesized mechanism of Ag CF formation and rupture. We have now inserted this discussion to the Supporting Information. Please refer to page 15 Supplementary Note 2 Supplementary Fig. 18 and page 9 main text.

Figure R1.8 Activation energy calculation. Double log I–V curves in the positive voltage sweeping region of the memristors provide insights on the charge transport mechanism in the **a** LRS and **b** HRS. **c** shows the calculation of the activation energy by studying the variation in the HRS as a function of temperature.

References:

- [1] Ilyas, N. et al. Analog switching and artificial synaptic behavior of Ag/SiO_x: Ag/TiO_x/p++-Si memristor device. **Nanoscale Res. Lett.** 15, 1–11 (2020).
- [2] Yoo, E. et al. Bifunctional resistive switching behavior in an organolead halide perovskite based Ag/CH₃NH₃PbI_{3-x}Cl_x/FTO structure. **J. Mater. Chem. C** 4, 7824–7830 (2016).
- [3] Duijnste, E. A. et al. Toward understanding space-charge limited current measurements on metal halide perovskites. **ACS Energy Lett.** 5, 376–384 (2020).
- [4] Le Corre, V. M. et al. Revealing Charge Carrier Mobility and Defect Densities in Metal Halide Perovskites via Space-Charge-Limited Current Measurements. **ACS Energy Lett.** 6, 1087–1094 (2021).
- [5] Lee, SangMyeong, et al. "Tailored 2D/3D halide perovskite heterointerface for substantially enhanced endurance in conducting bridge resistive switching memory." **ACS Applied Materials & Interfaces** 12.14 (2020): 17039-17045.

o Does it actually matter that these are CsPbBr₃ NCs or would the same effects occur for a CsPbBr₃ thin film? At such a thin layer (and given the level of detail of schematics in Figure 2), it would seem like a CsPbBr₃ thin film may work just as well.

We thank the Reviewer for the comment. As per the Reviewer's suggestion, we have carried out additional experiments to compare the performance of CsPbBr₃ NCs vs thin films. The results are presented in Figures R1.9 and R1.10 below. The thickness of the NC layer is increased to ~100nm by 5-6 serial spin coating steps. 100nm thick CsPbBr₃ thin films were spin coated (5000 rpm for 30 s) from equimolar precursor solutions of 0.25 M CsBr and 0.25 M PbBr₂ in DMSO solvent and annealed at 100 °C for 15 min under nitrogen environment to remove the solvent residue. The thicknesses of the other interlayers were fixed to: PEDOT:PSS- 30nm, pTPD- 20nm, and Ag- 150nm.

Both configurations exhibit very similar characteristics with an on-off ratio $\geq 10^3$, similar set and reset voltages. The volatile endurance performance is also similar and lasts 2 million cycles in both the thin film and NC film format (data not shown). However, stress tests of endurance and retention in the non-volatile mode reveal distinct responses. While CsPbBr₃ NCs depict high retention (10^5 secs) and endurance (5655 cycles) performance in the non-volatile mode, CsPbBr₃ thin films portray low retention (10^4 secs) and endurance (1174 cycles) performance in the non-volatile mode (Figure R1.9). These results point to different migration pathways for the Ag⁺ ion in NCs vs thin films.

Most importantly, we observe that CsPbBr₃ thin film memristors fail to be reconfigured back to their volatile mode once their non-volatile mode is activated, similar to other dual functional memristors reported in literature. On the other hand, the NC-based memristor allow facile reconfiguration between the volatile and non-volatile modes (Figure R1.10). We have now inserted this discussion to the Supporting Information. Please refer to pages 31-33 Supplementary Note 7 Supplementary Figs. 32-33 and page 15 main text.

Figure R1.9 Thin film vs NC film comparison. (reading top to bottom) I-V characteristics, retention and endurance performance of CsPbBr₃ thin film (left) and NC film (right) memristor in their non-volatile mode.

Figure R1.10 Switching between volatile and non-volatile modes on demand. **a** CsPbBr₃ thin film memristors are unable to switch to a volatile mode once the non-volatile mode is activated, but **b** the OGB-capped CsPbBr₃ NC memristors can facilitate switching between the 2 modes.

While the exact memristive mechanism is still unclear, our results favour NC memristor implementations empirically. Theoretically, it is plausible for a single nanocrystal to support multiple Ag filament formation-disruption processes. Since a single device encompasses 1000s of such nanocrystals, the NC layer provides a pool of electrochemically-active sites for filament formation. In comparison to thin films, where a specific migration path along grain boundaries may be preferentially opted, the NC layer provides a larger number of possibilities for forming CFs. Moreover, the ligands can further control the extent of electrochemical reactions as seen from our *in operando* thermal imaging experiments. Our work shows that larger insulating ligands may fair better in controlling the electrochemical reactions, hence resulting in higher endurance. But this is only a preliminary study and much more needs to be unravelled in future works and advances in halide perovskite nanocrystal chemistry will play a significant role in this regard.

o Why are pTPD and PEDOT:PSS chosen specifically? What role do they play in ion migration, charge transport, etc.?

The PEDOT:PSS + pTPD interlayers allow better perovskite NC thin film formation. The coating of CsPbBr₃ NC thin film directly on ITO results in clusters of NCs with poor adhesion. On PEDOT:PSS, the films are better when compared to ITO. But presence of large crevices once again indicate poor film formation of the NCs. Devices built with this configuration (ITO/ PEDOT:PSS/ CsPbBr₃ NC thin film/Ag) do not give us reliable resistive switching and multiple devices are short (data not shown). On the other hand, CsPbBr₃ NC thin film formation is best when coated on PEDOT:PSS + pTPD interlayers as evident from the AFM images (Figure R1.11). This is in alignment with similar observations on halide perovskite solar cells [Safari, Zeinab, et al. "Optimizing the interface between hole transporting material and nanocomposite for highly efficient perovskite solar cells." *Nanomaterials* 9.11 (2019): 1627.].

Figure R1.11. Role of PEDOT:PSS and pTPD interlayers. AFM images of CsPbBr₃ NC thin film on ITO (scan area = 2 μm × 2 μm), ITO + PEDOT:PSS (scan area = 20 μm × 20 μm) and ITO + PEDOT:PSS + pTPD interlayers (scan area = 20 μm × 20 μm).

We also conducted control experiments on ITO/PEDOT:PSS/Ag and ITO/PEDOT:PSS + pTPD/Ag structures to delineate the contribution of these interlayers to the memristive switching ability. As shown in Figure R1.12, these interlayers do not contribute to the resistive switching behaviour, reiterating the importance of the perovskite NC thin film as an active matrix for reliable and robust Ag filament formation and rupture. The devices could be set to a LRS, indicating that Ag can be pushed through these interlayers. But these devices could not be reset or erased back to their HRS. Devices switched reliably only with the presence of the CsPbBr₃ NC thin film as reported in Fig. 2 (main text). Thus, we conclude that perovskite NC thin film is the crucial active matrix that allows reliable and robust Ag filament formation and rupture. Both these processes are crucial for robust and reliable ‘write’ and ‘erase’ operations, resulting in high endurance. We have now inserted this to the Supporting Information. Please refer to pages 13-14 Supplementary Note 2, Supplementary Figs. 15-16 and page 9 main text.

Figure R1.12. IV of PEDOT:PSS and pTPD-only devices. IV sweep of ITO/PEDOT:PSS/Ag and ITO/PEDOT:PSS + pTPD/Ag devices.

o Does the silver have to be in direct contact with the perovskite NC layer for the devices to function? Do they function similar if a transport layer is inserted between the Ag electrode and the perovskite layer and can this help inform about the role of filament formation?

The silver does not have to be in direct contact with the perovskite NC layer for the devices to function. As per the Reviewer’s suggestion we spin coated a thin layer of PMMA (150nm) on top of the NC layer before Ag evaporation, i.e. ITO/PEDOT:PSS/pTPD/NC layer/PMMA/Ag. These devices exhibit very similar characteristics to the original PMMA-free configuration with an on-off ratio $\geq 10^3$, similar set and reset voltages Figure R1.13 [left panel]. Although PMMA encapsulates the NC layer from the ambient and allows longer storage shelf-life, statistical analysis of the on-off ratios and endurance (Figure R1.13 [right panel]) does not reveal any operational dependence on the PMMA coating.

Figure R1.13. IV sweep of ITO/PEDOT:PSS/pTPD/NC layer/PMMA/Ag device (left) and the intra-endurance statistics of on-off ratio (right).

Since the memristive mechanism in our devices is based on the migration of Ag^+ ions, we do not expect the insertion of a charge transport layer to modify the switching characteristics majorly other than possibly altering the magnitude of set and reset voltages. Transport layers such as spiro have been observed to allow migration of even noble species such as Au [Kerner, Ross A., et al. "Low Threshold Voltages Electrochemically Drive Gold Migration in Halide Perovskite Devices." *ACS Energy Letters* 5.11 (2020): 3352-3356.]. Hence, migration of Ag^+ ions through similar transport layers is feasible. However, we cannot envision how this would help us extract underlying mechanisms of filament formation. In fact, we believe that such a strategy would complicate the mechanistic analysis further because of the added possibilities of electrochemical reactions at the additional perovskite-transport layer and transport layer-metal interfaces.

Reviewer #2 (Remarks to the Author):

The authors have demonstrated a reconfigurable halide perovskite nanocrystal memristor that achieves on-demand switching between diffusive/volatile and drift/non-volatile modes. Moreover, the CsPbBr₃ nanocrystals capped with oleylguanidinium bromide (OGB) ligands exhibit remarkable endurance performances in both volatile and non-volatile modes. In addition, the authors constructed a fully-memristive reservoir computing framework with diffusive perovskite memristors as reservoir elements and drift perovskite memristors as readout elements. However, the following issues should be addressed.

1. The authors defined that the diffusive mode is volatile and the drift mode is nonvolatile. Please clearly tell the criteria to distinguish the volatile mode from the nonvolatile mode?

We thank the reviewer for suggesting to clarify the categorical difference of volatile and non-volatile mode functionality. The main difference of volatile and non-volatile modes is the duration that device can hold the information in its conductance levels. In our work, we followed the typical timescales used in the neuromorphic community when distinguishing both modes, i.e. 10-100 milliseconds retention for volatile behaviour [1-2], and $>10^3$ seconds retention for non-volatile behaviour [3-4]. The former is commonly used to implement short-term plasticity (STP), whereas the latter is used for the long-term plasticity (LTP) dynamics in synapses.

According to our measurements, in the drift-based non-volatile mode, memristor conductance stay stable until 10^5 seconds after programming (Fig. 2b, Supplementary Note 2, Supplementary Fig. 8). And in the diffusion-based volatile mode, each programming pulse induces a conductance increase of the device, which then decays back to original state in < 23 ms (Supplementary Note 3, Supplementary Fig. 23). Hence, the volatile mode is characterized by retention time in the range of tens of milliseconds, whereas the non-volatile mode depicts retention in the scale of hours. To make the distinction between volatile and non-volatile functionality of modes clearer to the reader, we have modified the following sentence in the manuscript.

"For example, the latest state-of-the-art spiking neural network (SNN) models require memory elements operating at multiple timescales, with both volatile and non-volatile properties (from tens of milliseconds to hours)". Please refer to page 2 in the main text.

[1] Zhu, X., Wang, Q. & Lu, W. D. Memristor networks for real-time neural activity analysis. *Nat. Commun.* 11, 1–9 (2020).

[2] Wang, Z. et al. Memristors with diffusive dynamics as synaptic emulators for neuromorphic computing. *Nat Mater* 16, 101–108 (2017).

[3] Ambrogio, S. et al. Equivalent-accuracy accelerated neural-network training using analogue memory. *Nature* 558, 60–67 (2018).

[4] Nandakumar, S. R. et al. Mixed-Precision Deep Learning Based on Computational Memory. *Front Neuroscience* 14, 406 (2020).

We have also modified the caption of Figure 2 to explain how endurance in the volatile and non-volatile modes were extracted for better clarity. Please refer to page 9 in the main text.

For endurance measurements in the volatile mode, the following methodology was used:

1. Read the current level of the device using +0.1V.
2. Apply +2V for 5ms as the write pulse and monitor the device's current level.
3. Repeat step 1.

Since our volatile memory loses the stored information upon removing power, the ON state is reported as the current value corresponding to the application of the programming pulse (at 2V) and the OFF state is reported as the current value corresponding to the application of the reading pulse (at 0.1V). This is in alignment with the methodology widely adopted in literature [e.g. Wang, Zhongrui, et al. "Memristors with

diffusive dynamics as synaptic emulators for neuromorphic computing." *Nature Materials* 16.1 (2017): 101-108.].

On the other hand, for endurance measurements in the non-volatile mode, the conventional methodology was used, i.e. the ON-OFF ratios are extracted from the current values corresponding to the same reading pulse (0.1V):

1. Read the current level of the device using +0.1V.
2. Apply +5V/-5V for 5ms as the write/erase pulses.
3. Repeat 1. Extract the on-off ratio comparing steps 1 and 3.

2. In Figure 2, the endurance measurements show an inferior non-volatile endurance than the volatile endurance. Please discuss the underlying mechanisms.

The operation of our perovskite devices depend on the migration of electrochemically active Ag^+ species through the perovskite matrix as explained in the manuscript. Here, the halide perovskite acts like a scaffold to enable this process. The formation and rupture of such conductive filaments (CFs) depend on a lot of factors. Among these factors, the applied electric field and Joule heating play the most important roles. Both these factors can play complimentary/competing roles and in determine the nature of the set/reset processes [1, 2].

By setting compliance currents (I_{cc}), we essentially set a limit to the extent of electrochemical reactions that occur during the migration of Ag^+ species through the perovskite matrix, in turn controlling the thickness of the CF or number of CFs to some extent. When the device is operated under a low I_{cc} of $1\mu\text{A}$, the filaments formed are thin and unstable and can dissolve spontaneously, resulting in a volatile memory. And since these processes have a low electrochemical and thermal budget, it becomes feasible to repeat the processes many times, resulting in high volatile endurance [3]. When the I_{cc} is raised to 1mA , the filaments formed are relatively thicker or more number of filaments can be formed, making it difficult to initiate the dissolution process. Hence, the devices preserve the CFs even when powered off, i.e. they are non-volatile. A large negative voltage is required to reset our bipolar devices and the Joule heating generated during this process ruptures the CFs. In the non-volatile mode, since these processes have a high electrochemical and thermal budget, it becomes difficult to repeat the processes many times, resulting in low non-volatile endurance [4]. This is a common observation in all memristive devices, irrespective of the active switching material as can be seen from Supplementary Note 5 Supplementary Table 2.

To investigate this further, we monitored the working of OGB memristors in their volatile and non-volatile modes *in operando* using a thermal camera. We observe that maximum heat is generated in our memristors during the reset process when trying to break the conductive filament(s). Application of reverse bias causes Joule heating which ruptures the conductive filament(s). Thick filaments are difficult to break and result in large rise in temperature in the surrounding areas which we pick up as an infrared image [5-7]. Figure R2.1 shows the thermograms recorded at the endurance limit of the respective modes – 2 million cycles for the volatile and 5000 cycles for the non-volatile mode. When the OGB device reaches its maximum non-volatile endurance/near failure (5000 cycles), an 84°C rise in surface temperature is noted. These findings indicate that an applied field alone cannot rupture the conductive filaments and that the reset process is likely to be a combined effect of electric field and Joule heating. However, no rise in surface temperature is observable after 2 million cycles of volatile endurance, supporting our hypothesis of better management of the electrochemical reactions with lower I_{cc} . As a result, the volatile endurance is much larger when compared to the non-volatile endurance. We have now inserted this discussion to the Supporting Information. Please refer to pages 27-28 Supplementary Note 6 Supplementary Fig. 30.

Figure R2.1 Thermal Camera Imaging. *In operando* monitoring of OGB memristors in their volatile and non-volatile modes using an infrared camera.

References:

- [1] Sun, Wen, et al. "Understanding memristive switching via in situ characterization and device modeling." *Nature Communications* 10.1 (2019): 1-13.
- [2] Kim, Sungho, Hee-Dong Kim, and Sung-Jin Choi. "Compact Two-State-Variable Second-Order Memristor Model." *Small* 12.24 (2016): 3320-3326.
- [3] Guo, M. Q., et al. "Unidirectional threshold resistive switching in Au/NiO/Nb: SrTiO₃ devices." *Applied Physics Letters* 110.23 (2017): 233504.
- [4] Chen, Yang Yin, et al. "Understanding of the endurance failure in scaled HfO₂-based 1T1R RRAM through vacancy mobility degradation." 2012 *International Electron Devices Meeting*. IEEE, 2012.
- [5] Gogoi, Himangshu Jyoti, and Arun Tej Mallajosyula. "Enhancing the Switching Performance of CH₃NH₃PbI₃ Memristors by the Control of Size and Characterization Parameters." *Advanced Electronic Materials* (2021): 2100472.
- [6] Li, Yibo, et al. "Review of memristor devices in neuromorphic computing: materials sciences and device challenges." *Journal of Physics D: Applied Physics* 51.50 (2018): 503002.
- [7] Yoon, Kyung Jean, et al. "Electrically-generated memristor based on inkjet printed silver nanoparticles." *Nanoscale Advances* 1.8 (2019): 2990-2998.

3. The I-V curve in Figure 2a only shows the positive voltage range. Does the I-V curve remain the same in the reverse voltage sweep?

Our device exhibits a unidirectional DC threshold switching behaviour (Figure R2.2) with no switching occurring under reverse bias (negative voltage on the Ag electrode). This can be correlated to the dominant bipolar electrode effect over thermal-driven diffusion, in alignment with literature [1-3].

Figure R2.2. Bidirectional IV sweep of OGB-capped CsPbBr₃ memristor in the volatile mode.

We have now inserted these details to the Supporting Information. Please refer to page 6 Supplementary Note 2, Supplementary Fig. 5 and page 7 main text.

References:

- [1] Midya, Rivu, et al. "Anatomy of Ag/Hafnia-based selectors with 1010 nonlinearity." **Advanced Materials** 29.12 (2017): 1604457.
- [2] Wang, Zhongrui, et al. "Threshold switching of Ag or Cu in dielectrics: materials, mechanism, and applications." **Advanced Functional Materials** 28.6 (2018): 1704862.
- [3] Guo, M. Q., et al. "Unidirectional threshold resistive switching in Au/NiO/Nb: SrTiO₃ devices." **Applied Physics Letters** 110.23 (2017): 233504.

4. In Figure 3b, there are 25 devices in the simulation of the reservoir computing. Should the uniformity of the 25 devices affect the accuracy of the training and test process? Does the uniformity needs to be considered.

We thank the reviewer for this question. As the reviewer points out, the uniformity of 25 devices should affect the training and testing accuracies. In short, our simulations already consider the non-uniformity of 25 devices.

The reservoir layer experiment in Figure 3b involves the application of 4975 different neural firing pattern inputs for each of the four types (tonic, bursting, irregular, adaptation) to the reservoir nodes for temporal processing of the input patterns. Unlike the previous work [1], where each device response was simulated without considering any device non-uniformities; in our work, we conducted 4975 (patterns) x 4 (classes) = 19,900 different measurements on 25 devices, physically. Since our results are based on extensive number of real device measurements, we believe that we already capture existing non-uniformities within 25 devices (both in space: device-to-device, and in time: cycle-to-cycle non-uniformities). Regarding the impact of device uniformities on the temporal processing, a recent work suggests that modest level of non-uniformities in neural processing units might enable more stable and robust training [2].

References:

- [1] Zhu, X., Wang, Q. & Lu, W. D. Memristor networks for real-time neural activity analysis. **Nature Communications** 11, 1–9 (2020).
- [2] Perez-Nieves, N., Leung, V. C. H., Dragotti, P. L. & Goodman, D. F. M. Neural heterogeneity promotes robust learning. **Nature Communications** 12, 5791 (2021).

5. The second paragraph is not well logically connected to the preceding text. Please improve the writing. We are not sure which specific paragraph the Reviewer is referring to here. We hope that with the extensive amount of additional data and explanation provided in this revision, the manuscript is of high quality and clear to readers.

6. Are both the diffusive and drift mechanisms essentially related to the movement of ions and vacancies under an electric field? The difference lies in the fact that one forms a conducting filament, which can remain in a low resistance state for a long time. This is the so-called non-volatile state. But the other does not form a conducting filament. The low resistance state will revert to the high resistance state in a short time. This is the so-called volatile state. Please make appropriate clarification in the manuscript.

We thank the Reviewer for the comment. Yes we believe that both the diffusive and drift mechanisms are related to the movement of ions and vacancies. From the IV characteristics, we see an abrupt jump in conductance for both a low compliance current (I_{cc}) of $1\mu A$ in the volatile mode and a high compliance current (I_{cc}) of $1mA$ in the non-volatile mode (Figure 2). This persuades us to believe that conductive filaments (CFs) are formed in both the volatile and non-volatile modes, albeit with contrasting number or size, in alignment with literature [1-3]. With a low I_{cc} , we expect the CFs formed to be thinner or fewer in number when compared to the non-volatile mode with a higher I_{cc} . We have illustrated this clearly in Figure 2 as well as Supplementary Note 2 Supplementary Figures 4 and 6 with the respective electrochemical reactions possible. We have now clarified this further in the Supporting Information. Please refer to page 8 Supplementary Note 2.

References:

- [1] Shi, Yuanyuan, et al. "Electronic synapses made of layered two-dimensional materials." **Nature Electronics** 1.8 (2018): 458-465.
- [2] Xiao, N. et al. Resistive random access memory cells with a bilayer TiO_2/SiO_x insulating stack for simultaneous filamentary and distributed resistive switching. **Advanced Functional Materials** 27, 1700384 (2017).
- [3] Sivan, Maheswari, et al. "All WSe_2 1T1R resistive RAM cell for future monolithic 3D embedded memory integration." **Nature Communications** 10.1 (2019): 1-12.

7. Please explain why the formation of the conductive filament is forming-free? How does the ICC exactly regulate the threshold switching ?

We believe the formation of the conductive filament is forming-free in our devices due to the following factors:

i. The low activation energy of migration of Ag^+ allows easy formation of conductive filaments. To calculate the activation energy, we investigate the variation in the high resistance state (HRS) in the voltage range 0.01-0.1V as a function of temperature using the Arrhenius equation $R = R_0 e^{-E_a/k_B T}$ (Figure R2.3). The extracted value of 0.208eV matches values reported in literature for Ag^+ migration [1], supporting the hypothesized mechanism of Ag CF formation and rupture.

Figure R2.3 shows the calculation of the activation energy by studying the variation in the HRS as a function of temperature.

ii. The soft lattice of the halide perovskite NCs facilitates easy diffusion of the mobile ions in comparison to the hard latticed oxide dielectrics used conventionally in memristors most of which require electroforming process to initiate resistive switching [2].

iii. The NCs provide a pool of available sites for redox reactions resulting in easier conductive filament (CF) formation.

Applying a compliance current (I_{cc}) sets a limit to the ion migration and extent of electrochemical reactions, in turn controlling the number or/and thickness of the CF to some extent. When the device is operated under a low I_{cc} of $1\mu\text{A}$, the filaments formed are thin and unstable and can dissolve spontaneously, resulting in threshold switching [3]. This is a common strategy adopted for all dual functional memristors (Supplementary Note 5 Supplementary Table 2). We have now inserted this discussion to the Supporting Information. Please refer to page 15 Supplementary Note 2 Supplementary Fig. 18.

References:

- [1] Lee, S. *et al.* Tailored 2D/3D Halide Perovskite Heterointerface for Substantially Enhanced Endurance in Conducting Bridge Resistive Switching Memory. **ACS Applied Materials and Interfaces** 12, 17039–17045 (2020).
- [2] Lai, Minliang, *et al.* "Intrinsic anion diffusivity in lead halide perovskites is facilitated by a soft lattice." **Proceedings of the National Academy of Sciences** 115.47 (2018): 11929-11934.
- [3] Guo, M. Q., *et al.* "Unidirectional threshold resistive switching in Au/NiO/Nb: SrTiO₃ devices." **Applied Physics Letters** 110.23 (2017): 233504.

8. Why are the current organic ligands chosen? What are the molecular structures of the two organic ligands? Why do the perovskite nanocrystals with the DAAB ligands exhibit far inferior properties compared to the perovskite nanocrystals of the OGB ligands? Is it because the length of the molecular chain does not have the same segregation effect on Ag^+ and Br^- ? The results of the PL measurements do not seem to be sufficient evidences for the vast difference in performance between the two ligands.

Didodecyldimethylammonium bromide molecule (DDAB) is a quaternary ammonium salt with two methyl groups and two carbon chains consisting of 12 carbon atoms, whereas oleylguanidinium bromide (OGB) is a guanidinium salt with one unsaturated chain containing 18 carbons. One of the most important advantages of OGB is the ability of the guanidinium group to create multiple hydrogen bonds with halides on the NC surface that can result in the better binding of such ligands to the surface of lead halide perovskite NCs. The second main advantage of OGB molecule is that the longer carbon chain improves colloidal stability of the NC solution.

Structure of the OGB (oleylguanidinium bromide) ligand:

Structure of the DDAB (didodecylammonium bromide) ligand:

It is known that commonly used ligands in the synthesis of lead halide perovskite NCs (oleic acid and oleylamine) loosely bind to the NC surface. This leads to the loss of colloidal stability and structural integrity of perovskite NCs. To overcome instability issues a new generation of organic ligands with various head groups should be developed and used. Therefore, we focused on the CsPbBr₃ NCs with quaternary ammonium and novel guanidinium-based long-chain ligands on the surface.

The way we can imagine the attachment of ligands to the cesium halide terminated surface of perovskite NCs is when some cesium ions are replaced by the head groups of the cationic ligands such as DDAB or OGB. It has already been shown (doi: [10.1021/acsenergylett.8b01669](https://doi.org/10.1021/acsenergylett.8b01669)) that DDAB ligands improve the

chemical stability of perovskite NCs and allow their purification preserving high photoluminescence quantum yield.

At the same time, we developed a synthesis of OGB-capped CsPbBr₃ NCs because we assume that guanidinium binding group in OGB molecule has a higher ability to create multiple hydrogen bonds with halides on the NC surface in comparison to DDAB molecule that can result in stronger binding to the CsPbBr₃ NCs.

Another difference between DDAB and OGB ligands is in their packing density on the surface of perovskite NCs. Carbon chains of DDAB are bulky and stand out which makes their fitting on the NC surface complicated. On the contrary, OGB molecule has only one oleyl chain with a double bond that, most probably, will allow packing of more ligands on the surface of perovskite NCs.

DDAB ligand has shorter carbon chains what makes them more useful in applications where charge transfer is important, such as LED devices. However, OGB has a longer carbon chain creating a bigger distance between NCs and making charge transfer harder. All these features described above can strongly impact the performance of devices based on DDAB or OGB-capped CsPbBr₃ NCs.

The dynamic ligand binding in halide perovskite (HP) nanocrystals (NCs) have been widely observed to create surface traps [1]. Therefore, handling the surface ligands without sacrificing the optoelectronic properties or affecting the structural integrity poses a challenge to developing semiconductive HP NC films. The large PL quenching observed with DDAB ligands (solution vs thin film comparison shown in Supplementary Fig. 12) point to creation of large number of surface traps. In perovskite films and NCs, it has been firmly established that trap states at the grain boundaries and on the surface can capture photo-excited charge carriers to create a local electric field capable of promoting ion migration [2-7]. In our case, this could lead to enhanced Ag⁺ and Br⁻ migration and eventually lead to thicker Ag filament formation. From our IV endurance measurements, the DDAB devices are observed to

- (i) quickly transit from a volatile to a non-volatile state, even at a low compliance current (I_{cc}) of 1 μ A resulting in an inferior volatile endurance of ~ 10 cycles, and
- (ii) quickly transit to a non-erasable non-volatile state at high I_{cc} of 1 mA, resulting in an inferior non-volatile endurance of ~ 50 cycles.

Both these results support the hypothesis of enhanced electrochemical reactions in the DDAB system due to their short chains.

To investigate this further, we monitored the working of DDAB and OGB memristors *in operando* using a thermal camera. We observe that maximum heat is generated in our memristors during the reset process when trying to break the conductive filament(s). Application of reverse bias causes Joule heating which ruptures the conductive filament(s). Thick filaments are difficult to break and result in large rise in temperature in the surrounding areas which we pick up as an infrared image [8-10]. Figure R2.4 shows the thermograms recorded at the time of failure of the memristors- 50 cycles for DDAB and 5655 cycles for OGB. For the OGB device, a thermogram is also recorded at the 50th endurance cycle for direct comparison to the DDAB device. At the time of failure of the DDAB device, i.e. 50th endurance cycle, a 95°C rise in surface temperature is seen. In comparison, only a 13°C rise is observed for OGB devices at the 50th cycle. When the OGB device reaches its maximum endurance/near failure (5655 cycles), a 86°C rise in surface temperature is noted similar to the DDAB device. These findings indicate that an applied field alone cannot rupture the conductive filaments and that the reset process is likely to be a combined effect of electric field and Joule heating. Hence it is critical to engineer materials to regulate the underlying electrochemical reactions. This experiment further supports our hypothesis of better management of the electrochemical reactions with the larger OGB ligands when compared to DDAB, and points to the importance of investigating nanocrystal-ligand chemistry for the development of high-performance robust memristors.

We have now inserted this discussion to the Supporting Information. We have explained the rationale for ligand choice on pages 2-3 Supplementary Note 1 Supplementary Fig. 1 and detailed the thermal imaging results on pages 29-30 Supplementary Note 6 Supplementary Fig. 31.

References:

1. Xue, Jingjing, Rui Wang, and Yang Yang. "The surface of halide perovskites from nano to bulk." **Nature Reviews Materials** 5.11 (2020): 809-827.
2. DeQuilettes, Dane W., et al. "Photo-induced halide redistribution in organic–inorganic perovskite films." **Nature Communications** 7.1 (2016): 1-9.
3. Azpiroz, Jon M., et al. "Defect migration in methylammonium lead iodide and its role in perovskite solar cell operation." **Energy & Environmental Science** 8.7 (2015): 2118-2127.
4. Ahn, Namyong, et al. "Trapped charge-driven degradation of perovskite solar cells." **Nature Communications** 7.1 (2016): 1-9.
5. Kim, Young-Hoon, et al. "Charge carrier recombination and ion migration in metal-halide perovskite nanoparticle films for efficient light-emitting diodes." **Nano Energy** 52 (2018): 329-335.
6. Kamat, Prashant V., and Masaru Kuno. "Halide ion migration in perovskite nanocrystals and nanostructures." **Accounts of Chemical Research** 54.3 (2021): 520-531.
7. Chen, Mingming, et al. "Manipulating ion migration for highly stable light-emitting diodes with single-crystalline organometal halide perovskite microplatelets." **ACS Nano** 11.6 (2017): 6312-6318.
8. Gogoi, Himangshu Jyoti, and Arun Tej Mallajosyula. "Enhancing the Switching Performance of CH₃NH₃PbI₃ Memristors by the Control of Size and Characterization Parameters." **Advanced Electronic Materials** (2021): 2100472.
9. Li, Yibo, et al. "Review of memristor devices in neuromorphic computing: materials sciences and device challenges." **Journal of Physics D: Applied Physics** 51.50 (2018): 503002.
10. Yoon, Kyung Jean, et al. "Electrically-generated memristor based on inkjet printed silver nanoparticles." **Nanoscale Advances** 1.8 (2019): 2990-2998.

Figure R2.4 Thermal imaging. *In operando* monitoring of the DDAB and OGB memristors using an infrared camera.

9. What is the delayed system? Please give the definition of separability in this system? How does this relate to the characteristics of the device?

We thank the reviewer for these questions. The term delay system corresponds to a non-linear system with delayed feedback and/or a delayed coupling, introduced by Appeltant, L. et al. [1]. One simple example of delay systems is a single non-linear node whose dynamics is affected by its own output a time t in the past. Thus, delay systems can be easily implemented by combining a single nonlinear node and a delay loop. For better clarity, we have modified the caption of Figure 1 where the term delayed system is being used, to include the definition of delay systems provided by [1]. Please refer to page 5 in the main text.

In our work, similar to the methodology used in previous research [1, 2], we employed the delay system concept to implement a reservoir computing framework. We explained how our perovskite material can be used as a nonlinear node (please see the section on Diffusive perovskite memristors as reservoir elements, page 10 main text) and how we implemented the delay loop (section Classification of neural firing patterns, page 14 main text) in the context of delay systems. Briefly, as we apply the input patterns to a single perovskite memristor, the memristor conductance evolves due to internal nonlinear device dynamics, and the resulting change in the conductance states is recorded with a fixed sampling period. Because each measurement is influenced by both the current input and the recent device states, this system acts as a delayed system.

The term separability on classification tasks with the reservoir networks refers to a degree of which the reservoir states resulting from different class of inputs can be clustered by linear hyperplanes [3]. Please see Figure 2 from Appeltant, L. et al. for illustration of linear separability. In the delayed systems, the input signal is projected into a high-dimensional reservoir space by using the time-domain recordings of a dynamically excited, physical, nonlinear device. And during the readout training procedure with the gradient descent algorithm, linear hyperplanes (represented by the weights of the readout layer) are positioned to better separate the high-dimensional reservoir states belonging to different classes. To provide reader with a direct reference on separability, we have now cited Gibbons's 2010 paper where the separation property in the reservoir computing is mathematically described and further discussed. Please refer to page 10 in the main text.

According to [1], a reservoir network needs to satisfy two properties for efficiently solving classification tasks. 1) It needs to project the input signal to a high-dimensional state; and the higher the dimension, the more likely that the data becomes linearly separable [4]. 2) The dynamics of reservoir should exhibit a non-linear short-term memory (being influenced by inputs from recent past, but not from the far past) for a better representation. First requirement is not related to the characteristics of the device but relates to how many times it is sampled during the excitation. For the second requirement, we extensively investigated and verified the non-linearity of our perovskite device as follows:

- We analyzed the non-linear behavior of the device as a function of applied electrical pulse. We altered the pulse amplitude, pulse width and pulse number to observe the non-linear conductance change. See Supplementary Note 3, Supplementary Fig. 22, page 19.
- We analyzed the non-linear, short-term property of the device by applying different temporal sequences and monitored the evolution of the conductance. See Supplementary Note 3, Supplementary Fig. 23, page 20.
- We analyzed the separability of different pattern sequences by monitoring the final device conductances. See Supplementary Note 3, Supplementary Fig. 24, page 21.

References:

[1] Appeltant, L. et al. Information processing using a single dynamical node as complex system. **Nature Communications** 2, 468 (2011).

- [2] Zhu, X., Wang, Q. & Lu, W. D. Memristor networks for real-time neural activity analysis. **Nature Communications** 11, 2439 (2020).
- [3] Gibbons, T.E., 2010, July. Unifying quality metrics for reservoir networks. In *The 2010 International Joint Conference on Neural Networks (IJCNN)* (pp. 1-7). IEEE.
- [4] Cover, T. M. Geometrical and Statistical Properties of Systems of Linear Inequalities with Applications in Pattern Recognition. **IEEE Transactions on Electronic Computers** 3 (1965): 326-334.

10. Why is the conducting filament described by using the word “thick”? Why can a thick filament not be erased by negative bias?

We use the terms ‘thin’ and ‘thick’ filaments to figuratively explain/illustrate the extent of filament formation within the devices. This is in alignment with the terminologies used in literature such as **Nature Materials** 16.1 (2017): 101-108., **Nature Communications** 10.1 (2019): 1-12., **Advanced Electronic Materials** 7.2 (2021): 2000866., **Applied Physics Letters** 115.14 (2019): 143501.. Intuitively, thicker or larger conductive filaments (CFs) can form permanent ‘short circuits’ within the device stack which can be difficult to erase. These are also often associated with runaway electrochemical reactions that can change the nanoelectrochemical structure of the material, and can cause permanent breakdown of the active matrix [1-3].

References:

- [1] Kim, Tae-Hyeon, et al. "Fabrication and characterization of TiO_x memristor for synaptic device application." **IEEE Transactions on Nanotechnology** 19 (2020): 475-480.
- [2] Abbas, Yawar, et al. "Compliance-free, digital SET and analog RESET synaptic characteristics of sub-tantalum oxide based neuromorphic device." **Scientific Reports** 8.1 (2018): 1-10.
- [3] Simanjuntak, Firman Mangasa, Takeo Ohno, and Seiji Samukawa. "Film-nanostructure-controlled inerasable-to-erasable switching transition in ZnO-based transparent memristor devices: sputtering-pressure dependency." **ACS Applied Electronic Materials** 1.11 (2019): 2184-2189.

REVIEWERS' COMMENTS

Reviewer #1 (Remarks to the Author):

The authors have addressed the reviewer comments and questions.

Reviewer #2 (Remarks to the Author):

The authors have largely addressed the previous comments. Now the manuscript is clearly improved. This work has demonstrated an important advance for the applications of semiconductor nanocrystals. I believe that researchers in the field of semiconductor nanocrystals may be quite interested in this work. In this context, the authors should consider to at least extend the introduction of this manuscript by referring to literature on the exploration of semiconductor nanocrystals for neuromorphic computing (e.g., ACS Appl. Mater. Interfaces 2021, 13, 32, <https://doi.org/10.1021/acsami.1c06096>; Nano Energy 52 (2018) 422–430 <https://doi.org/10.1016/j.nanoen.2018.08.018>), attracting broader readership.

Reviewer #1 (Remarks to the Author):

The authors have addressed the reviewer comments and questions.

We thank the Referee for approving the revised changes and endorsing our work.

Reviewer #2 (Remarks to the Author):

The authors have largely addressed the previous comments. Now the manuscript is clearly improved. This work has demonstrated an important advance for the applications of semiconductor nanocrystals. I believe that researchers in the field of semiconductor nanocrystals may be quite interested in this work. In this context, the authors should consider to at least extend the introduction of this manuscript by referring to literature on the exploration of semiconductor nanocrystals for neuromorphic computing (e. g., ACS Appl. Mater. Interfaces 2021, 13, 32, <https://doi.org/10.1021/acsami.1c06096>; Nano Energy 52 (2018) 422–430 <https://doi.org/10.1016/j.nanoen.2018.08.018>), attracting broader readership.

We thank the Referee for the positive comments on our work and the revised changes.

As per the Referee's suggestion, we have now expanded the introduction section to describe the exploration of semiconductor nanocrystals for neuromorphic computing and have included the above-mentioned references. Please refer to page 4 in the main text, also reproduced below.

“NCs in general are recently garnering significant attention for artificial synaptic implementations because they support a wide range of switching physics such as trapping and release of photogenerated carriers at dangling bonds over a broad spectral region⁴⁰, and single-electron tunnelling⁴¹. They allow low-energy (<fJ), high-speed (MHz) operation, and can support scalable and CMOS-compatible fabrication processes. In the case of perovskite NCs, however, existing implementations often utilize NCs only as a charge trapping medium to modulate the resistance states of another semiconductor, in flash-like configurations a.k.a synaptic transistor^{42–45}. The memristive switching capabilities and limits of the perovskite NC active matrix remains unaddressed, entailing significant research in this direction. Colloids of perovskite nanocrystals (NCs) are readily processable into thin-film NC solids and they offer a modular approach to impart mesoscale structures and electronic interfaces, tunable by adjusting the NC composition, size and surface ligand capping.”